# Local unfolding of the HSP27 monomer regulates chaperone activity

T. Reid Alderson [1,2], Julien Roche [2,3], Heidi Y. Gastall[1], David M. Dias[1], Iva Pritišanac[1], Jinfa Ying[2], Ad Bax[2], Justin L.P. Benesch [1] & Andrew J. Baldwin [1]

The small heat-shock protein HSP27 is a redox-sensitive molecular chaperone that is expressed throughout the human body. Here, we describe redox-induced changes to the structure, dynamics, and function of HSP27 and its conserved α-crystallin domain (ACD). While HSP27 assembles into oligomers, we show that the monomers formed upon reduction are highly active chaperones in vitro, but are susceptible to self-aggregation. By using relaxation dispersion and high-pressure nuclear magnetic resonance (NMR) spectroscopy, we observe that the pair of β-strands that mediate dimerisation partially unfold in the monomer. We note that numerous HSP27 mutations associated with inherited neuropathies cluster to this dynamic region. High levels of sequence conservation in ACDs from mammalian sHSPs suggest that the exposed, disordered interface present in free monomers or oligomeric subunits may be a general, functional feature of sHSPs.

[1] Department of Chemistry, Physical and Theoretical Chemistry Laboratory, University of Oxford, South Parks Road, Oxford OX1 3QZ, UK. [2] Laboratory of Chemical Physics, National Institute of Diabetes and Digestive and Kidney Diseases, National Institutes of Health, Bethesda, MD 20892, USA. [3] Present address: Roy J. Carver Department of Biochemistry, Biophysics and Molecular Biology, Iowa State University, Ames, IA 50011, USA. Correspondence and requests for materials should be addressed to J.L.P.B. (email: justin.benesch@chem.ox.ac.uk) or to A.J.B. (email: andrew.baldwin@chem.ox.ac.uk)

Small heat-shock proteins (sHSPs) are a class of molecular chaperones present in all kingdoms of life and exhibit diverse functionality, from modulating protein aggregation to maintaining cytoskeletal integrity and regulating apoptosis[1]. The most abundant sHSP in humans[2], HSP27 (or HSPB1), is systemically expressed under basal conditions and upregulated by oxidative stress[3], during aging[4], and in cancers[5] and protein deposition diseases[6]. Numerous mutations in HSP27 have been linked to different neuropathies, including distal hereditary motor neuropathy (dHMN) and Charcot–Marie–Tooth (CMT) disease[7,8], the most commonly inherited neuromuscular disorder. These maladies are themselves linked to oxidative stress[9,10], and recent studies have indicated that the reducing environment of the cytosol progressively transitions to an oxidising environment over the lifetime of an organism[11,12].

HSP27 is directly sensitive to the intracellular redox state via its lone cysteine residue (C137), which controls dimerisation by forming an intermolecular disulphide bond in vivo even under the reducing conditions of the cytosol[13]. This cysteine is highly conserved in HSP27 orthologs but not found in other mammalian sHSPs[14], implying that it plays an important functional role. Accordingly, the presence of this disulphide bond impacts on the activity of HSP27 in vitro[15–17] and on the resistance of cells to oxidative stress[13,14,18,19]. Like other mammalian sHSPs, HSP27 assembles to form a wide range of oligomers[20] whose constituent monomers and dimers freely exchange between oligomers[21,22]. The chaperone activities of many sHSPs have been characterised in vitro[23], but the active sHSP species remains unclear, with large oligomers[24,25], small oligomers[26,27], and dimers[28] all implicated. Intriguingly, variants of HSP27 that have an increased tendency to form free monomers display hyperactivity both in vitro and in vivo[29,30].

Although functionally relevant, no sHSP monomer has yet been characterised at atomic resolution, as they are typically present at low abundance in equilibrium with higher-order oligomers. Obtaining high-resolution structural information on HSP27 is challenging, as it assembles into a polydisperse ensemble of inter-converting oligomers ranging from approximately 12 to 36 subunits[31–33] of average molecular mass of ca. 500 kDa. Removal of the C-terminal region (CTR) and N-terminal domain (NTD) leaves a conserved ~80-residue, α-crystallin domain (ACD) that does not assemble beyond a dimer (Fig. 1a). The subunits in the dimer adopt an immunoglobulin-like fold, and assemble through the formation of an extended β-sheet upon pairwise association of their β6 + 7 strands[34–37]. Under oxidising conditions, the dimer interface in HSP27 is reinforced by an intermolecular disulphide bond involving C137 from adjacent subunits centred on a two-fold axis[34–36]. Based on evidence from the closely related human sHSP paralog, αB-crystallin (HSPB5)[38], the ACD is likely structurally similar in the context of the full-length oligomeric protein and in its isolated dimeric form. The excised ACD of both αB-crystallin and HSP27 can display potent chaperone activity in vitro[35,39], suggesting that important aspects of sHSP function are encoded within this domain.

Here, we have employed NMR and native mass spectrometry to interrogate the impact of redox-induced changes to the structural features of HSP27, its excised ACD (cHSP27), and mutants (C137S and H124K/C137S) that affect its ability to dimerise. Against a range of client proteins including citrate synthase (CS), malate dehydrogenase (MDH), α-lactalbumin (αLac), insulin, glyceraldehyde-3-phosphate dehydrogenase (GAPDH), and Parkinson's-disease-related α-synuclein (αS), we find that HSP27 is a more active chaperone under conditions that favour the release of free monomers. Under all conditions tested, the monomeric form of the ACD is a more effective

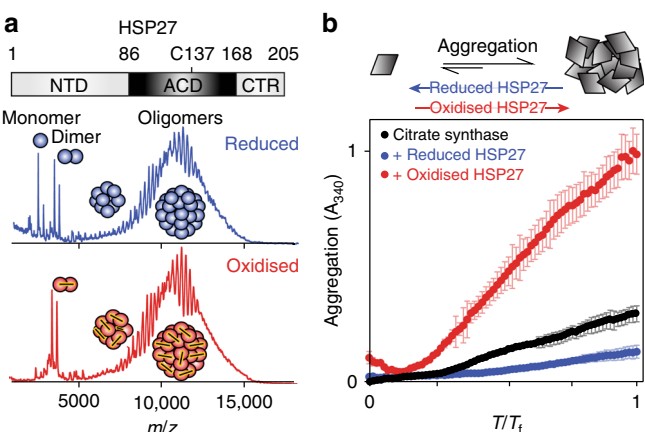

**Fig. 1** Reduction of HSP27 releases highly active monomers. **a** Domain architecture of the human molecular chaperone HSP27, which forms polydisperse oligomers that reach >500 kDa. Native mass spectra collected at 25 μM total monomer concentration for both oxidised (red) and reduced (blue, +250 μM dithiothreitol, DTT) HSP27 reveal the formation of polydisperse oligomers. More monomers are present in the reduced sample, although the oligomeric distributions are highly similar (Supplementary Fig. 1). **b** The chaperone activity of HSP27 was assayed by monitoring the increase in light scattering at 340 nm of 10 μM of CS in the presence or absence of 0.5 μM reduced (blue, 5 mM 2-mercaptoethanol, BME) or oxidised (red) HSP27. At this concentration, dimeric HSP27 comprises more than half of all populated stoichiometries. The average of two replicates is shown with error bars corresponding to ±1 standard deviation (SD)

chaperone than its dimeric counterpart, and against αLac the monomeric ACD almost entirely recapitulates the activity of the full-length chaperone. We demonstrate that neither the oligomeric distribution of HSP27, nor the structures or fast dynamics (ps–ns) of the cHSP27 dimer and disordered CTR vary appreciably with redox changes. Taken together, we conclude that an altered structure of the monomeric form is responsible for the redox dependent chaperone activity.

To interrogate the structure of the transiently populated monomers, we have used a combination of Carr–Purcell–Meiboom–Gill (CPMG) relaxation dispersion (RD) and high-pressure solution-state nuclear magnetic resonance (NMR) spectroscopy methods. Our data reveal that monomeric cHSP27 becomes highly dynamic and disordered in the region that previously constituted the dimer interface. While we find the cHSP27 monomer to be highly chaperone-active in vitro, we demonstrate that increasing the abundance of the monomer results in a heightened tendency for uncontrolled self-aggregation. The importance of the unstructured region in this delicate balance between function and malfunction can be linked to mutations in HSP27 that are associated with hereditary neuropathies, which mainly cluster to the disordered region of the monomer.

## Results

**Monomers from reduced HSP27 are highly chaperone-active.** We first examined full-length HSP27 to analyse redox-dependent changes to its oligomeric distribution. Native mass spectra of reduced and oxidised HSP27 were highly similar, with over-lapping signals in the 5000–15,000 m/z region (Fig. 1a), consistent with previous data[31,32]. This reveals that HSP27 assembles into large, polydisperse oligomers with similar oligomeric distributions under both conditions (Supplementary Fig. 1). We also

observed monomeric and dimeric HSP27 in the spectra of both oxidised and reduced forms, with a significant increase in the population of free monomer upon reduction (Fig. 1a).

To confirm that dissociation of the dimers, rather than modulation of the oligomers, is the major consequence of reduction, we used NMR to examine the CTR, which can mediate the assembly of sHSPs[40,41]. As HSP27 oligomers have an average mass of ca. 500 kDa, only the disordered CTR from [15]N-labelled HSP27 can be observed in a 2D [1]H-[15]N heteronuclear single quantum coherence (HSQC) NMR spectrum[42–44]. To probe the local dynamics in this region quantitatively, we recorded NMR spin relaxation experiments that characterise motions on the ps-ns timescale (Supplementary Fig. 1). No significant differences in the conformations or fast backbone motions were detected between oxidised and reduced forms of HSP27. Our combined native MS and NMR data on the polydisperse ensemble populated by HSP27 demonstrate that the primary impact of reduction is the release of free monomers.

To ascertain whether the presence of monomers impacts on chaperone function, we initially used the model substrate citrate synthase (CS)[23,45] to probe the activity of HSP27 in vitro. The addition of 0.5 μM reduced HSP27 significantly suppressed aggregation, a result that contrasts with oxidised HSP27, which appeared to co-aggregate with CS and enhance aggregation (Fig. 1b). Other sHSPs have been found to co-aggregate with substrates[46–48]. The oligomerization of HSP27 is concentration-dependent[32,39], and at ca. 2 μM total concentration 50% of the populated stoichiometries are expected to be dimers when oxidised[32]. At 0.5 μM, the equilibrium is further shifted to favour dimers, and thus we expect the majority of HSP27 to be present either as dimers (oxidised) or a mixture of non-covalent dimers and monomers (reduced, Fig. 1).[32] Our chaperone activity data suggest that monomerisation regulates the chaperone activity of HSP27, rendering it more effective at suppressing aggregation in vitro.

To test the generality of this result, we examined the ability of HSP27 to suppress aggregation for a range of aggregating proteins including αS and thermo-sensitive clients MDH and GAPDH (Fig. 2). Aggregation curves of MDH and GAPDH were independent of the addition of DTT, whereas amyloid formation

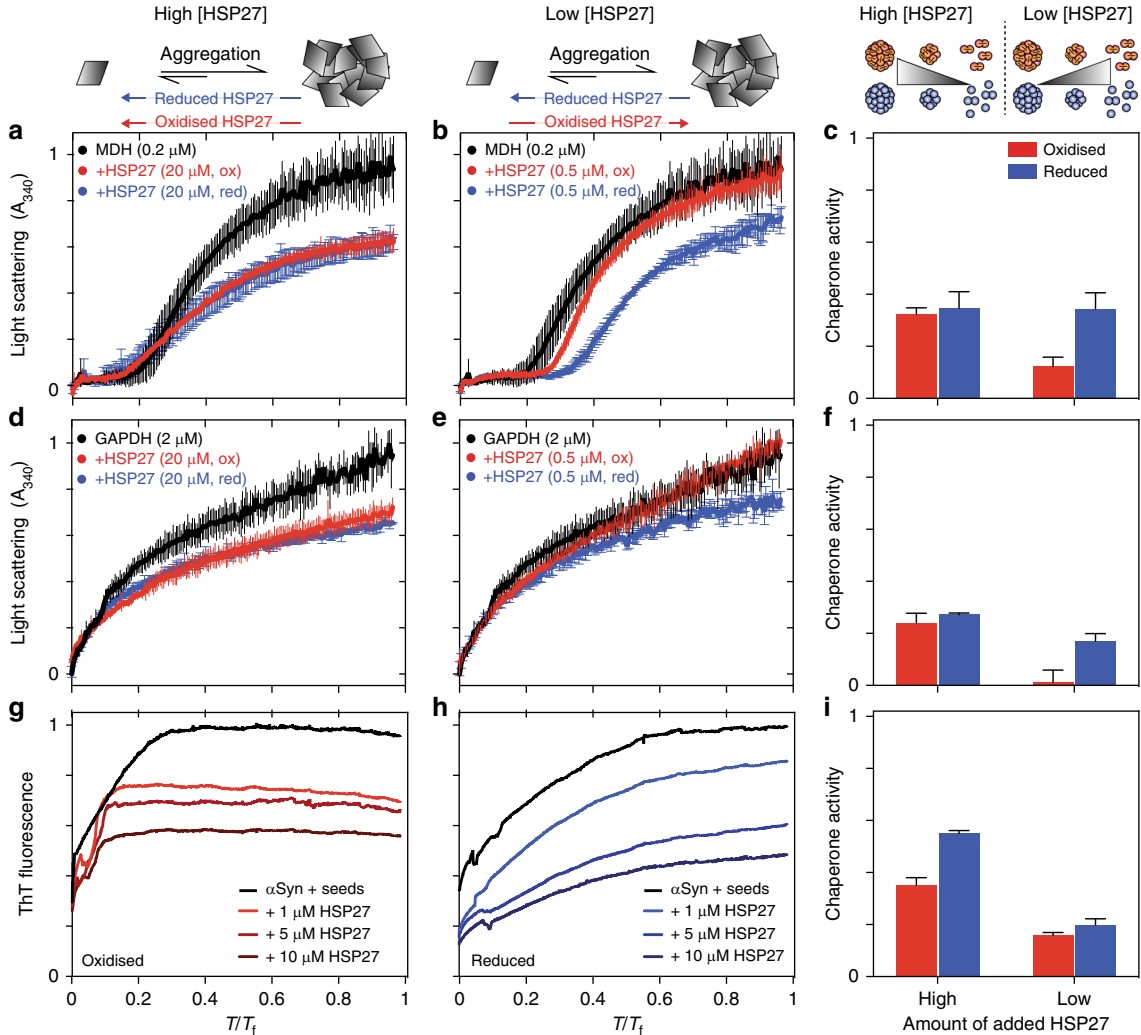

**Fig. 2** HSP27 monomers are potent chaperones in vitro. The aggregation of MDH (**a**, **b**) and GAPDH (**d**, **e**) was monitored by light scattering at 340 nm and seeded amyloid fibril formation by αS (**g**, **h**) was monitored by ThT fluorescence. Reduced (blue) or oxidised (red) HSP27 was added at either high (≥10 μM) concentrations (**a**, **d**, **g**) where it is predominantly oligomeric or low (≤1 μM) concentrations (**b**, **e**, **h**) where a large fraction of dimers (oxidised) or monomers (reduced) are populated. The y axes were normalised for comparison, and the x axes were scaled by a factor $T_f$ to normalise time. The $T_f$ values used for each substrate were 5 (**a**, **b**), 5 (**d**, **e**), and 170 (**g**, **h**). The chaperone activity of HSP27 against each substrate (**c**, **f**, **i**) is quantified as one minus the ratio of average signal over the time course with and without chaperone (Methods). Values of one and zero would respectively represent the complete inhibition of aggregation and no protection against aggregation. The average of three replicates is shown with error bars corresponding to ±1 SD

by αS was approximately 2.5-fold faster (Supplementary Fig. 2). We note that normalisation of the αS data suggests that the mechanism of aggregation was accelerated, but not altered by redox changes, thus allowing us to qualitatively compare chaperone activity. Strikingly, while the activity of HSP27 depends on the specific aggregating protein under study, HSP27 is a more effective chaperone under conditions that favour the release of free monomers.

**HSP27 monomers are potent chaperones that readily aggregate.** Given the potent chaperone activity of the HSP27 monomer, we sought to characterise the monomer:dimer equilibrium in more detail. To isolate this equilibrium from higher-order oligomer assembly, we turned to truncated forms that contains only the ACD, termed cHSP27 (Fig. 3a, Supplementary Fig. 3), which forms dimers whose structures are essentially independent of oxidation state[34–36]. In addition to the wild-type cHSP27 sequence, we produced two disulphide-incompetent variants, C137S and H124K/C137S[49]. The additional H124K mutation was introduced following prior observations that determined mimicking auto-protonation of the H124 side-chain destabilises the dimer[49]. Native MS at 5 μM revealed pure dimers (oxidised), pure monomers (H124K/C137S), or a mixture (C137S, reduced) (Fig. 3b). This redox-dependent monomerisation is consistent with observations in full-length HSP27 (Fig. 1a).

To directly compare the chaperone activity of the cHSP27 dimer and monomer, we assayed the ability of C137S and H124K/C137S to suppress the aggregation of α-lactalbumin (αLac)[35] and insulin[32], whose aggregation is initiated by the addition of DTT (Fig. 3b). For both aggregating clients, the monomeric form of cHSP27 (H124K/C137S) is a more active chaperone (Fig. 3b). Reduced cHSP27 and C137S prevented aggregation nearly identically, confirming the similarity between the two dimeric forms. We also compared the activity of cHSP27 to full-length HSP27, which was more potent than cHSP27 against prevent insulin aggregation, but the two forms were similar against αLac. These results further support that HSP27 exhibits client-dependent chaperone activity[23], and that, while the ACD can be an active chaperone, other components such as the NTD are also important[32]. Nevertheless, our results show that, against these clients, the monomeric ACD is more active than the dimer.

Interestingly, at elevated concentration (800 μM) and neutral pH, the H124K/C137S monomer showed a greater propensity than C137S to self-aggregate. H124K/C137S forms large amorphous aggregates (Fig. 3f, Supplementary Fig. 4) that did not display the Thioflavin T (ThT) binding characteristic of amyloid fibrils (Supplementary Fig. 4). The melting temperatures of both reduced cHSP27 and C137S diminished markedly with total concentration (Supplementary Fig. 4), indicative of lower thermodynamic stability upon monomerisation. Taken together, these results suggest that, in addition to being more active, the cHSP27 monomer is also kinetically unstable and aggregation-prone. We note that self-aggregation of H124K/C137S was not evident during the chaperone activity assays (Fig. 3) due to the usage of low concentrations.

**Dynamics at the ACD dimer interface are redox-sensitive.** To obtain insight into the structural rearrangements that trigger the enhanced chaperone activity and aggregation propensity of the cHSP27 monomer, we turned to NMR. 2D $^1$H-$^{15}$N HSQC NMR spectra of oxidised and reduced cHSP27 were recorded at 1 mM, a concentration that favours dimer formation. The NMR spectra of cHSP27 in both redox states were highly similar, a finding that is consistent with the 2.5-Å backbone RMSD between the two

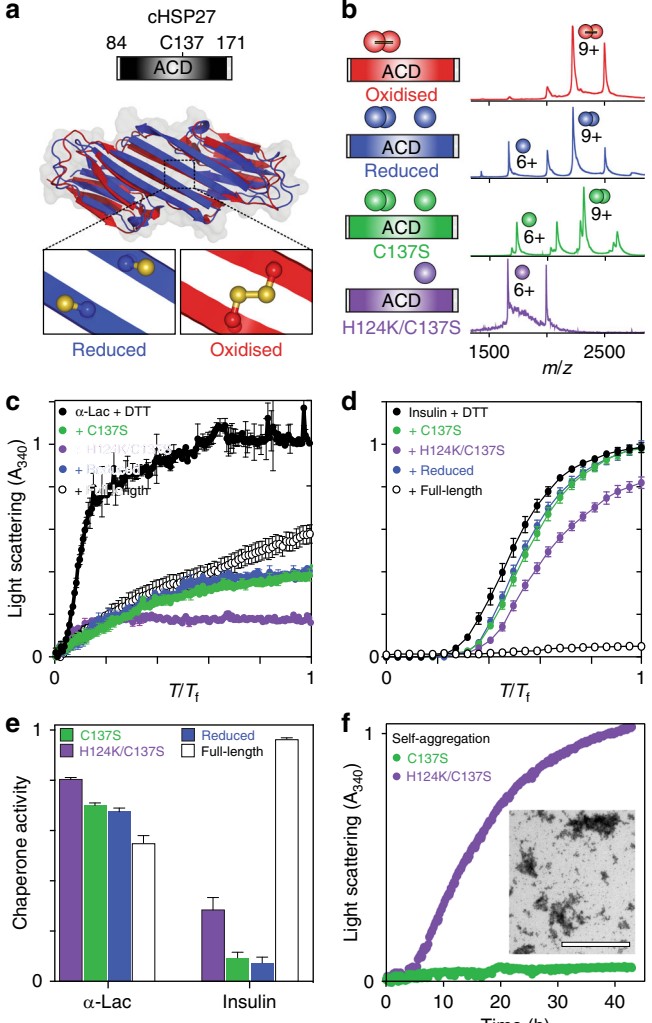

**Fig. 3** cHSP27 monomers are potent chaperones in vitro. **a** The central ACD exists as a stable dimer in which residue C137 at the dimer interface can access both reduced (blue, PDB 4mjh) and oxidised (red, PDB 2n3j) states. **b** The cHSP27 variants used in this study together with their native mass spectra at 5 μM. For reduced cHSP27, 250 μM DTT was added. **c** Aggregation of 300 μM α-lactalbumin (αLac; black) at 37 °C and **d** 80 μM insulin at 40 °C following the addition of 2 mM DTT, both monitored by light scattering at 340 nm. The values of $T_f$ are 10 (**c**) and 0.3 (**d**). Aggregation of both proteins is inhibited by C137S (green), reduced cHSP27 (blue), H124K/C137S (purple), and full length HSP27 (empty). For αLac and insulin, the sHSP concentrations were 70 and 40 μM, respectively. The traces represent the average of three experiments with errors bars indicating ± 1 SD. Under these conditions, C137S and reduced cHSP27 are predominantly dimeric whereas H124K/C137S is monomeric **e** Relative chaperone activity of the cHSP27 variants from panels **c** and **d**, with activity defined as one minus the ratio of the average signal with and without chaperone. **f** The self-aggregation of 800 μM C137S and H124K/C137S monitored by light scattering at 340 nm at 37 °C. The traces represent the average of six (C137S) or three (H124K/C137S) experiments with error bars indicating ± 1 SD. Inset: A representative transmission electron microscopy (TEM) micrograph at 40 h reveals large, non-fibrillar aggregates of H124K/C137S

forms (Fig. 4a). The NMR data confirmed that both the secondary structure and hydrogen-bonding network (Supplementary Fig. 3) in our construct were consistent with published structures[34–36]. Moreover, $^{15}$N relaxation experiments revealed that

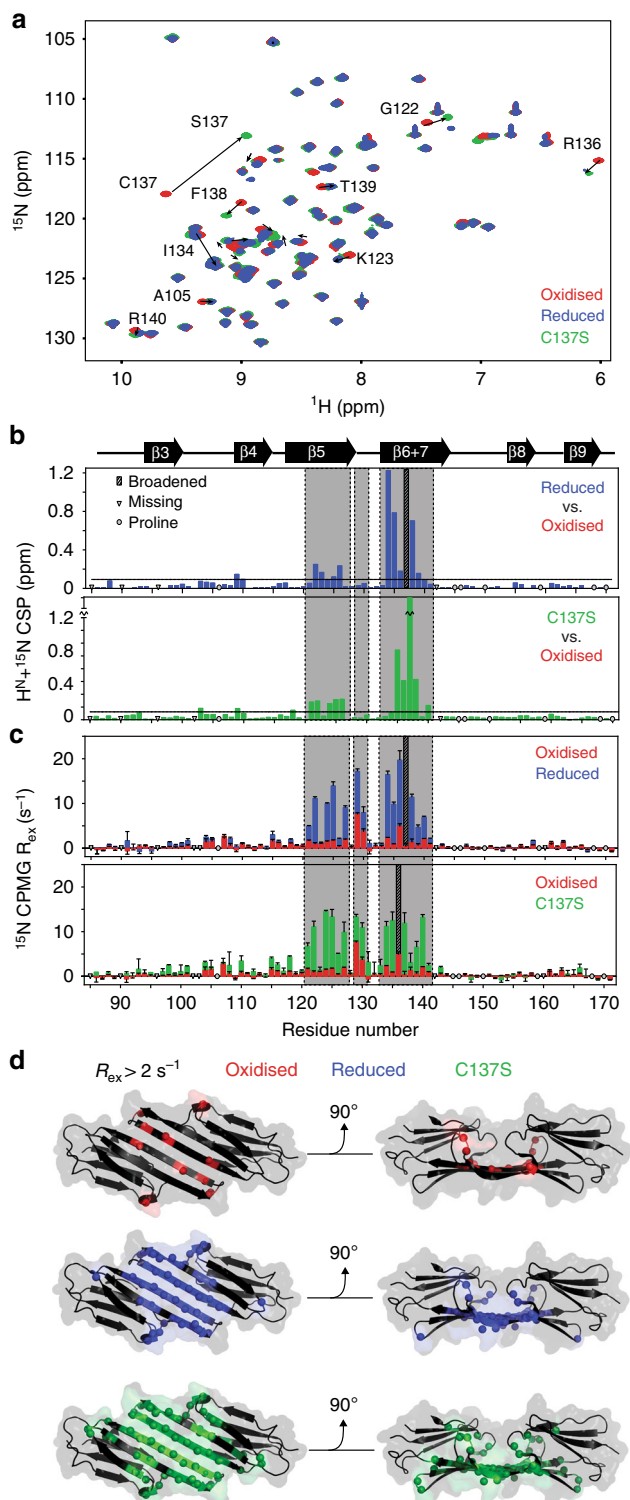

**Fig. 4** Redox-induced perturbations to the structure and dynamics of cHSP27. **a** Overlaid 2D $^1H$-$^{15}N$ HSQC spectra of oxidised (red), reduced (blue, +5 mM BME), and C137S (green) cHSP27 reveal their structural similarities. Significant CSPs are indicated with arrows. **b** CSPs for individual residues between reduced (top) or C137S (bottom) and the oxidised state reveal differences that localise to the β5 and β6 + 7 strands. The resonance from C137 was broadened beyond detection in reduced cHSP27 (cyan). Proline residues that do not contribute to this spectrum and unassigned residues are indicated. CSPs were calculated as $\sqrt{(\Delta H^2 + (0.2\Delta N^2))}$. **c** $R_{ex}$ versus residue number, which qualitatively probes μs–ms motions. Errors bars are 1 SD, as derived from propagation of the fitted errors in measured $R_{2,eff}$ values. Residues near $L_{5,6+7}$ show large $R_{ex}$ values in oxidised, reduced, and C137S cHSP27. The motion extends into the β5 and β6 + 7 strands for reduced cHSP27 and C137S. **d** $R_{ex}$ values > 2 s$^{-1}$ are mapped onto the structure of cHSP27 (PDB 4mjh), indicating that residues with slow dynamics cluster near the dimer interface

Fig. 3), with elevated $^{15}N$ transverse relaxation rates (Supplementary Fig. 5). Reduction of the disulphide bond therefore leads to dynamics on the μs-ms timescale near the dimer interface of cHSP27 that attenuate NMR signals. Conformational fluctuations between multiple states on this timescale can be characterised using CPMG RD NMR spectroscopy experiments[50,51], which employ a variable pulse frequency, $\nu_{CPMG}$, to measure effective transverse ($R_2$) relaxation rates[52–54]. The relative populations of the states that are interconverting ($p_G$, $p_E$), their rate of interconversion ($k_{ex}$), and the chemical shift differences ($|\Delta\omega|$) associated with the structural changes can be obtained through quantitative analysis of CPMG RD data.

With $^{15}N$ CPMG RD, we observed motions on the μs–ms timescale in the β5, $L_{5,6+7}$, and β6 + 7 regions of reduced cHSP27 and C137S, with oxidised cHSP27 only exhibiting motions in $L_{5,6+7}$ and near the disulphide bond (Fig. 4c, d, Supplementary Fig. 6, Supplementary Table 1, Supplementary Table 2, Supplementary Table 3, Supplementary Table 4, Supplementary Table 5). H124K/C137S was insufficiently stable to perform the CPMG RD measurements (Fig. 3f). To allow us to characterise the motions in detail, we collected an extensive set of CPMG RD data on C137S spanning multiple temperatures (25, 30, and 35 °C), concentrations (0.3 and 1 mM), and magnetic field strengths (11.7 and 14.1 T). The motions could be interpreted in terms of a global, two-state exchange between the principally populated dimer and a sparsely populated, but partially disordered monomeric state (Fig. 5c, Supplementary Fig. 6, Supplementary Table 1, Supplementary Table 2). From the CPMG RD data, we calculated a $K_d$ for the C137S monomer–dimer equilibrium of 0.5 μM (Supplementary Fig. 6, Supplementary Table 1), a result qualitatively consistent with results obtained by native MS (Fig. 3b). Activation and thermodynamic parameters of the monomer-dimer inter-conversion were obtained by analysing the variation in CPMG RD data with temperature. Dissociation of the dimer was endothermic, consistent with a disruption of stabilising interactions, and entropically favoured, consistent with an increase in structural disorder (Supplementary Table 1). The transition state for dissociation was more disordered than the dimer as evidenced by a positive activation enthalpy and entropy (Supplementary Fig. 6, Supplementary Table 1).

Similar to reduced cHSP27 and C137S, oxidised cHSP27 exhibited μs–ms dynamics in $L_{5,6+7}$ (Fig. 5a, Supplementary Fig. 6, Supplementary Table 3), but not in β5. The motions in $L_{5,6+7}$ were found to involve unfolding of the loop, thereby disrupting the intermolecular salt bridge between D129 in $L_{5,6+7}$ and R140 in β6 + 7 from the adjacent monomer (Fig. 5b, Supplementary Table 3). In addition to the local unfolding of $L_{5,6+7}$, the oxidised

the ps-ns backbone dynamics were essentially unaltered by changes in redox state (Supplementary Fig. 5). Similarly, we confirmed that C137S mimics the reduced form, as their NMR spectra revealed very similar CSPs to the oxidised state (Fig. 4b), apart for the residues immediately adjacent to the C137S mutation.

Although the structure of the underlying dimer[35,36] and fast backbone dynamics were redox-independent, NMR signal intensities for residues in the vicinity of the reduced and C137S dimer interfaces were substantially attenuated (Supplementary

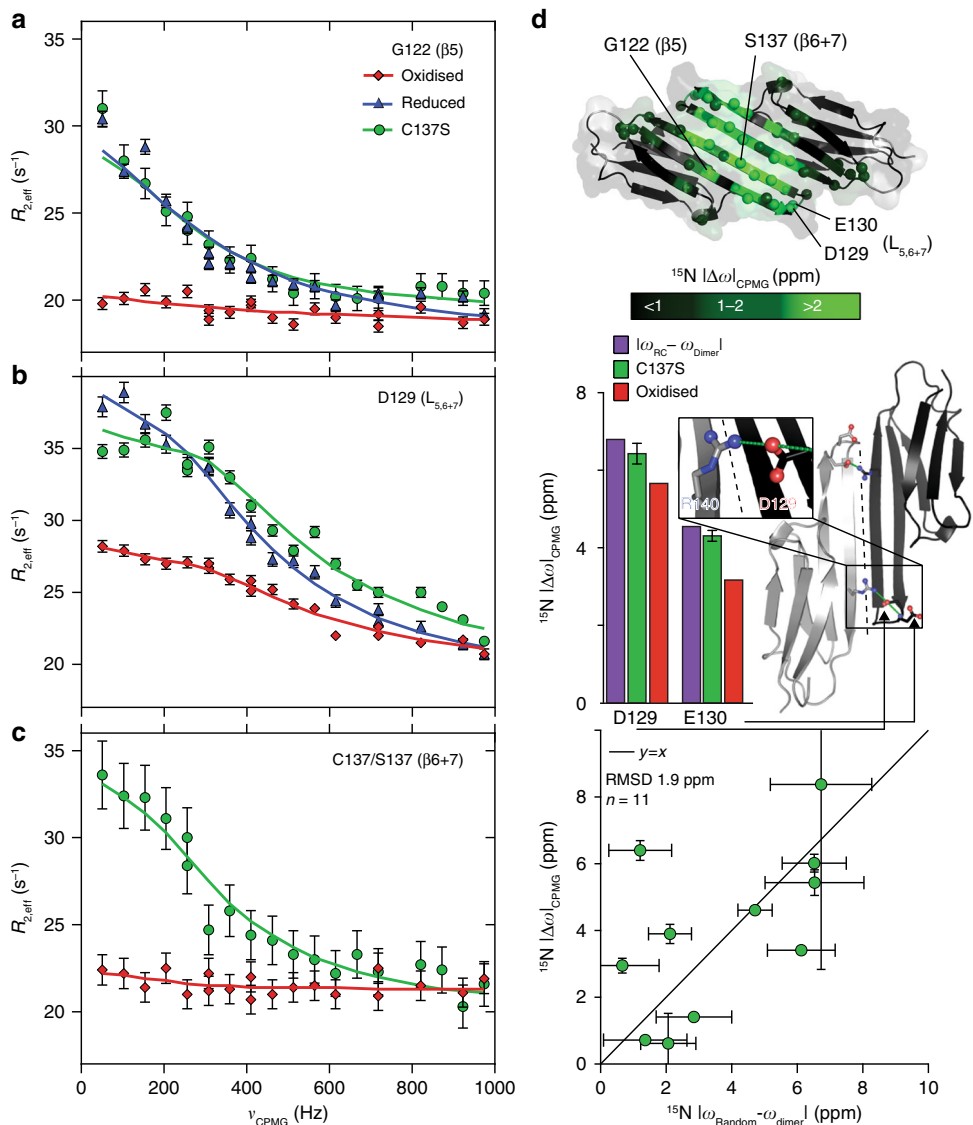

**Fig. 5** Relaxation dispersion reveals partial unfolding of the cHSP27 monomer. $^{15}$N CPMG RD experiments quantify µs–ms motions in oxidised (red), reduced (blue), and C137S (green) cHSP27. Fitted curves from a global analysis are shown as solid lines. Significant CPMG RD curves were observed in the β5 strand (**a**), $L_{5,6+7}$ loop (**b**), and the β6 + 7 strand (**c**). Redox-independent motions were observed in $L_{5,6+7}$, which arise from unfolding of the loop, whereas only the non-covalent dimers in C137S and reduced cHSP27 show motions throughout β5 and β6 + 7. **d** (top) CPMG RD-derived $^{15}$N chemical shift changes in C137S (|Δω|) plotted onto the structure (PDB: 4mjh) reveal structural changes in $L_{5,6+7}$ and the β5 and β6 + 7 strands. (middle) The $^{15}$N | Δω| values in $L_{5,6+7}$ are similar in both C137S and oxidised cHSP27, and correlate with those expected for a transition to a random coil, indicating that unfolding of $L_{5,6+7}$ is independent of oxidation state. In $L_{5,6+7}$, D129 forms an intermolecular salt bridge with R140 from an adjacent subunit, and the amide nitrogen from E130 forms a hydrogen bond with the carbonyl of D129 within the same subunit. All error bars are derived from fitting and represent SD values. (bottom) $^{15}$N |Δω| values in $L_{5,6+7}$ and β6 + 7 upon monomerisation are compared to the changes expected for random coil formation. The agreement is reasonable, indicating the monomer is substantially disordered in these regions. Error bars represent SD values

form of cHSP27 showed µs–ms motions in the vicinity of residue C137 on a faster timescale, consistent with isomerisation of the disulphide bond[47] (Supplementary Fig. 6, Supplementary Table 4).

**Partially disordered monomers characterised by RD NMR.** Given the increased chaperone activity of the monomeric ACD (Fig. 3b, c), we pursued a structural characterisation of the C137S monomer. While unable to observe the C137S monomer directly, the $^{15}$N chemical shift differences obtained from the CPMG RD experiments (|Δω|) report on the structure of the monomeric state. The |Δω| values that we obtained indicate that residues at the dimer interface (β6 + 7, $L_{5,6+7}$) adopt random-coil-like disordered conformations[48] (Fig. 5c, Supplementary Fig. 8, Supplementary

Table 2). Consistent with this finding, we observed that the H124K/C137S monomer displayed characteristics of a partially disordered protein, as evidenced by its 2D $^1$H-$^{15}$N HSQC spectrum (Supplementary Fig. 4), circular dichroism spectrum, and bis-ANS (4,4′-dianilino-1,1′-binaphthyl-5,5′-disulfonic acid) fluorescence. While most of the molecule retains its fold upon monomer release, as noted by small |Δω| values (Supplementary Fig. 8), residues become disordered in $L_{5,6+7}$ and β6 + 7, the dimer interface.

**Direct detection of the monomer at high pressures and low pH.** While CPMG RD enables a direct characterisation of the HSP27 monomer under near-physiological conditions, its low population (ca. 1.5% at 1 mM) renders further high-resolution analysis

challenging. We thus sought to stabilise the monomeric fold to directly observe it by NMR. A well-resolved resonance (G116) provided a straightforward marker for distinguishing between the monomeric and dimeric states. Two resonances from this residue were observed in slow exchange at low concentrations for reduced cHSP27 and C137S, and the $^{15}N$ $|\Delta\omega|$ between the two resonances (1 ppm) matched the value obtained from the CPMG analysis (1.1 ppm). From these intensities, we determined a $K_d$ for C137S of 0.7 μM, a value consistent with the CPMG analysis (Supplementary Table 1). Similarly, the concentration dependence of the intensity of the G116 monomer and dimer resonances for H124K/C137S revealed an increase in the $K_d$ by 3 orders of magnitude to ca. 1.1 mM.

To preserve the monomer at high concentrations for characterisation by NMR spectroscopy, increased hydrostatic pressure[55] was employed. NMR spectra of C137S as a function of pressure were recorded at pH 7 in a baroresistant buffer[56] whose pH does not vary with pressure. These spectra revealed a shift in the equilibrium from folded dimer at low pressure to entirely unfolded monomeric C137S at high pressure (Fig. 6a, Supplementary Fig. 7) via an intermediate species that was maximally populated at 1600 bar (Fig. 6b, c, Supplementary Fig. 7). The variation in populations of dimer, monomer, and unfolded monomer as a function of pressure were explained quantitatively by a three-state linear equilibrium model (Fig. 6b, Supplementary Table 1). Volumetric changes upon application of pressure were obtained, together with the equilibrium constant of monomer unfolding, $K_u$, at 1 bar (Supplementary Table 1), revealing a free energy difference ($\Delta_u G$) of $5 \pm 0.4$ kJ mol$^{-1}$ between the monomer and unfolded species at 1 bar and pH 7. The $K_d$ for dimerisation increased ten-fold at 1600 bar (Fig. 6c, Supplementary Fig. 7), and was further increased by three orders of magnitude upon lowering from pH 7 to 5 at 1 bar (Fig. 6a, Supplementary Fig. 7, 8). By contrast, no significant change in the spectrum of oxidised cHSP27 at low pH was observed, consistent with its adoption of a rigid dimer (Supplementary Fig. 7). We were able to combine the effects using a phosphate buffer whose pH decreases with pressure, to maximally stabilise the monomeric form of C137S (Fig. 6b).

While the C137S monomer aggregated under acidic conditions at elevated protein concentrations, it remained stable up to 100 μM at pH 4.1 at 1 bar. The monomeric nature of this sample was confirmed by NMR translational diffusion measurements (Supplementary Fig. 8). Triple resonance spectra of the monomer were acquired under these conditions (Fig. 7a). All observable $H^N$, N, Cα, and CO nuclei were assigned (Fig. 7a, Supplementary Fig. 8) and, similar to observations by CPMG RD, the largest CSPs relative to the C137S dimer fell in $L_{5,6+7}$ and $\beta 6 + 7$ (Fig. 7b, Supplementary Fig. 8, 9). A reasonable correlation was observed (RMSD 1.6 ppm, Supplementary Fig. 8) when the $^{15}N$ chemical shift differences from CPMG RD acquired at pH 7 were compared to those measured directly at pH 4.1, indicating that the monomeric conformation is similar in both cases. Further confirming their structural similarity despite a nearly 3-unit change in pH, minor resonances from the monomeric protein could be observed in a sample of C137S at 20 μM at pH 7, and the observed monomeric chemical shifts were close to the values obtained directly under acidic conditions (Fig. 7a, Supplementary Fig. 8). We transferred the majority of resonance assignments from the C137S monomer at pH 4.1 to the H124K/C137S monomer at pH 7 (Supplementary Fig. 8). The CSPs between the C137S dimer and H124K/C137S monomer were consistent with those determined for the dimer-to-monomer transition at low pH (Supplementary Fig. 8), further confirming that the H124K/C137S monomeric variant resembles the C137S monomer at low and neutral pH.

**Structural and dynamical characterisation of the monomer.** To characterise the monomeric state of C137S, we used the observed chemical shifts to determine β-strand formation in the C137S dimer and monomer using the chemical shift[58] and random coil[57] indices (CSI, RCI, Fig. 7b, Supplementary Fig. 9). This analysis confirmed that, while the disordered $L_{5,6+7}$ spans from Q128 to Q132 in the dimer, it becomes substantially elongated in the monomer and includes residues between K123 and S137, thereby shortening the β5 and $\beta 6 + 7$ strands. Notably, the second half of the β5 and first half of the $\beta 6 + 7$ strands are not formed in the monomer, implying that these regions fold upon dimerisation.

Finally, we recorded {$^1$H}-$^{15}$N heteronuclear nuclear Overhauser enhancements (hetNOEs) for the monomer and dimer, which allowed a direct comparison of fast backbone motions on the ps-ns timescale[59]. In the monomer, residues in $L_{5,6+7}$, the C-terminal portion of β5, and the N-terminal portion of $\beta 6 + 7$ were highly dynamic (Fig. 7b, Supplementary Fig. 9), consistent with the CPMG RD data and chemical shifts (Fig. 7a). The rigid dimer interface in cHSP27 therefore partially unfolds and becomes highly dynamic in the monomeric state.

## Discussion

The function and monomerisation of the molecular chaperone HSP27 is regulated by its redox state through an inter-dimer disulphide bond (Figs. 1–3)[14,16,17]. Here, we determined the structural basis for this regulation. Remarkably, the oligomeric distribution and ps-ns dynamics within the flexible CTR in the context of full length HSP27, and both the structure ps-ns dynamics of cHSP27 dimers were invariant to formation of the disulphide bond in the dimer interface. Reduction of full-length HSP27 and cHSP27 leads predominantly to the release of the free monomers (Fig. 1b). We observe enhanced chaperone activity in vitro against multiple aggregating proteins when we increase the quantity of free monomers (Figs. 1–3), which was achieved by manipulating the full-length HSP27 concentration to favour monomer release and by comparing monomeric and dimeric ACDs of HSP27. We note that the core domains do not entirely recapitulate the activity of the full-length chaperone. In the case of insulin, the activity of the full-length chaperone was significantly greater than the ACDs, likely suggesting an important role for the NTD of the protein in chaperone activity[32], whereas against αLac, both monomeric and dimeric ACDs were more efficient chaperones than the full-length protein. Although in general, the specific activities of HSP27 and cHSP27 vary depending on the specific choice of aggregating protein[23], from direct comparisons between the monomeric ACD mutant H124K/C137S and the dimeric ACD (C137S and wild-type) (Fig. 3), we conclude that the monomeric ACD is a significantly more effective chaperone than the dimer against the aggregating proteins presented in this work.

Using a combination of CPMG RD and high-pressure NMR, we established that the monomeric ACD of HSP27 partially unfolds upon dissociation (Fig. 7), such that the region responsible for the rigid interface in the dimer becomes highly dynamic. We can attribute the difference in activity between the monomeric and dimeric forms to this structural rearrangement. The link between heightened disorder and enhanced activity have been previously observed for αB-crystallin (ABC). Under acidic conditions (pH 2.5), ABC exists as a predominantly unfolded monomer, where it can prevent the aggregation of β2-microglobulin[60]. Moreover, unstructured peptides from both ABC and αA-crystallin (AAC), including an 8-mer comprising the $\beta 6 + 7$ strand, are able to prevent substrate aggregation[61]. Consistent with these observations, both the acid-induced unfolding of HdeA and HdeB[62] and the oxidation-dependent

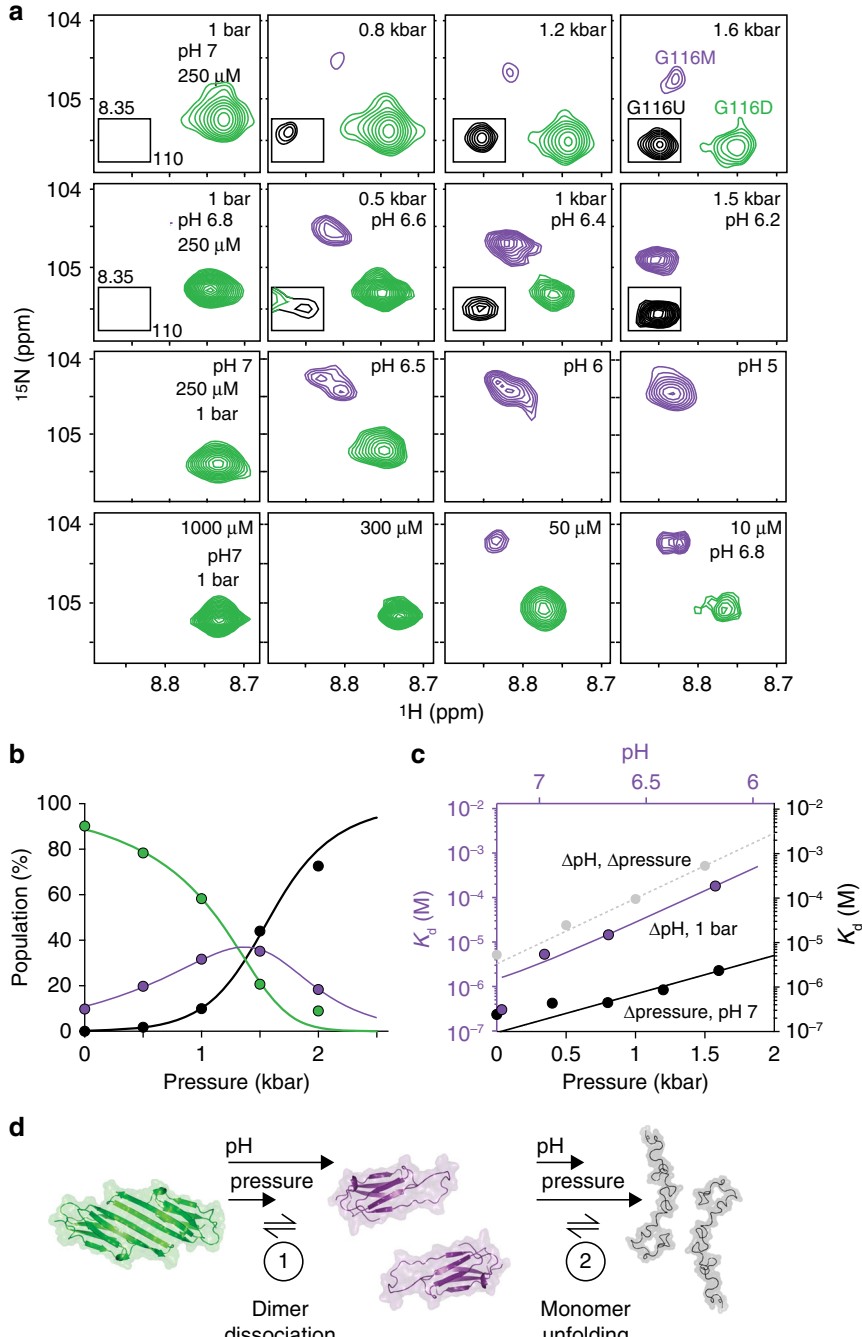

**Fig. 6** High pressure and low pH stabilise the cHSP27 monomer. **a** NMR spectra of C137S, focusing on residue G116 as a function hydrostatic pressure, pH, and concentration. Shown here are four rows of spectra: increasing pressures at pH 7 in a baro resistant buffer, increasing pressures at pH 6.8 at 1 bar in phosphate buffer (pH decreases with pressure), decreasing pH at 1 bar, and decreasing protein concentration at pH 7 at 1 bar. Resonances from the dimer (green), monomer (purple), and unfolded (black) state are readily distinguishable. Decreasing concentration or pH and increasing pressure favour the monomerisation. At pressures greater than ~1.5 kbar, the unfolded form becomes the principally populated state. **b** Variations in NMR signal intensities with pressure from four residues that were unambiguously assigned in all three conformations were well explained (solid lines) by the quantitative 3-state equilibrium model shown. **c** The $K_d$ for dimerisation is shown as a function of pressure (black), pH (purple), or their combination (grey). **d** The three-state mechanism of C137S dimer dissociation, where low pH and high pressure favour the partially disordered monomer, before ultimately the monomers unfold

unfolding of HSP33[63] result in potent molecular chaperones. More generally, the plastic nature of intrinsically disordered proteins (IDPs) is thought to aid their ability to bind a wide variety of partners via specific, yet transient interactions. It is possible that the same mechanism for rapid, promiscuous recognition of binding partners by IDPs is responsible for the heightened activity of partially unfolded chaperones. Recent reports have indicated that HSP27 interacts with substrates α-synuclein and Tau in a dynamic, transient manner[64,65] with CSPs induced by Tau binding identified in the β6 + 7 strand of cHSP27[65]. Interestingly, many of the residues that are unfolded in the HSP27 monomer are charged or polar (Fig. 7d), suggesting that electrostatics may play a significant role in substrate-recognition[66], in addition to hydrophobic interactions[67].

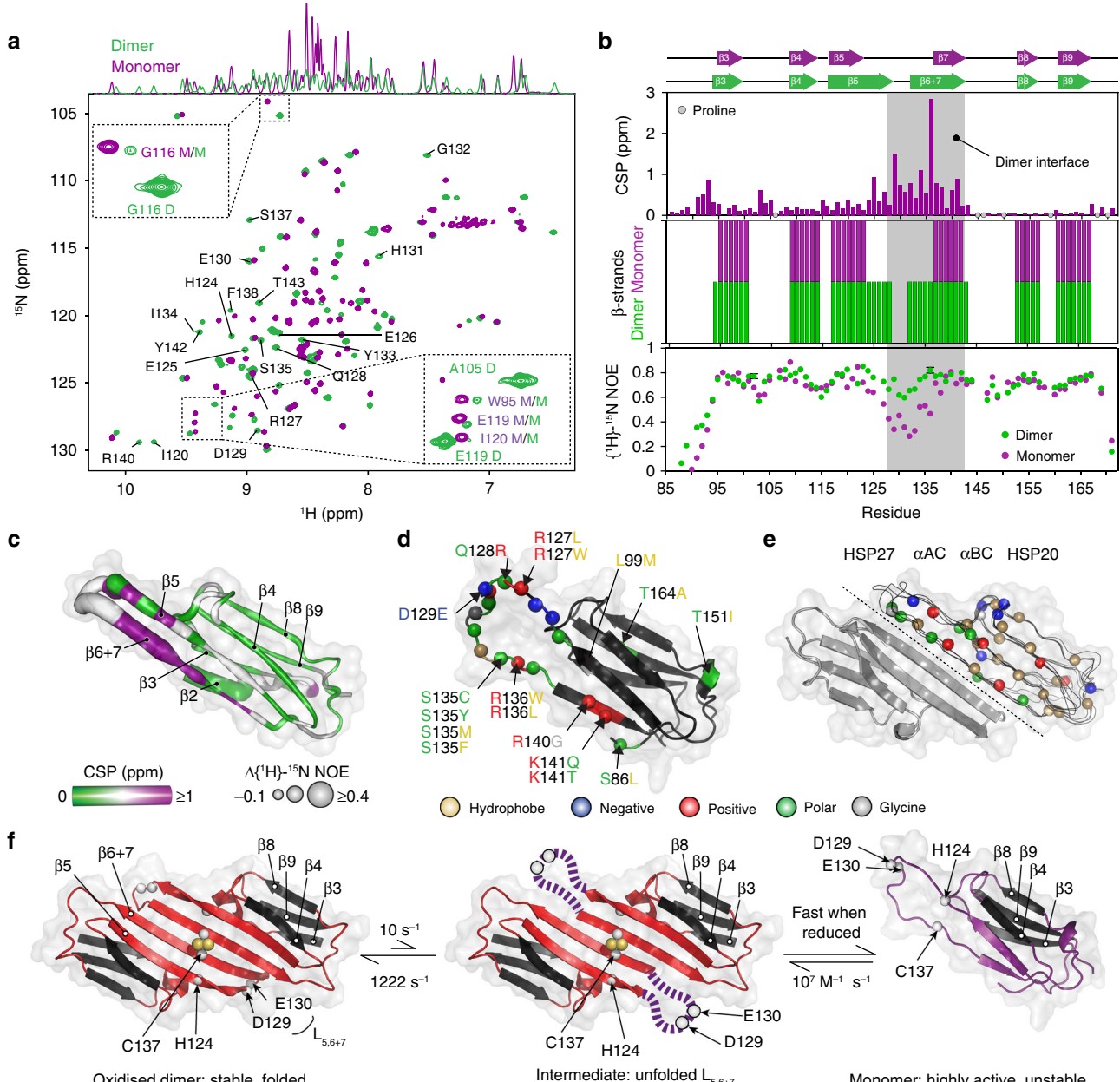

**Fig. 7** Structural and dynamical characterisation of the cHSP27 monomer. **a** Overlaid NMR spectra of C137S at 20 μM and pH 7 (green) where it is predominantly dimeric or pH 4.5 (purple) where it is monomeric. The 1D projection onto the $^{1}$H dimension reveals significant disorder in the monomer. Insets: minor resonances observable in the pH 7 sample correspond to the directly observed monomer species at pH 4.5. D/M corresponds to dimer/monomer, respectively. **b** (top) Combined and weighted $^{1}$H, $^{15}$N chemical shift perturbations (CSPs) between the C137S dimer at pH 7 and monomer at pH 4.5, with CSPs computed as described above. (middle) β-strands in the C137S monomer (purple) and dimer (green) as identified by RCI[57]. (bottom) {$^{1}$H}-$^{15}$N NOEs (hetNOEs) for the C137S dimer and monomer (Supplementary Fig. 9). Error bars are derived from signal-to-noise. **c** The ACD monomeric fold from PDB 4mjh is shown with the difference (dimer-monomer) in {$^{1}$H}-$^{15}$N NOE values (tube thickness) and magnitude of CSPs (colour). **d** Inherited mutations that are implicated in the onset of CMT or dHMN disease are indicated in the ACD of HSP27. The mutations cluster to the regions that become solvent exposed upon monomer formation and tend to lower the charge density in the region. **e** Overlaid dimer structures of human HSP27 (PDB 4mjh), human αB-crystallin (PDB 4m5s), bovine αA-crystallin (PDB 3l1f), and rat HSP20 (PDB 2wj5) are shown in ribbon format for one subunit of each dimer. Their similarity indicates the highly-conserved fold of vertebrate ACDs. The second subunit of HSP27 is shown in cartoon format. Highly conserved residues among human sHSPs (HSPB1-HSPB6) are shown as spheres with the same colour format as **d**. **f** Possible hierarchical mechanism of monomer formation. The oxidised, reduced, and C137S forms of cHSP27 exhibit similar dynamics in $L_{5,6+7}$ and form a disordered loop. In the absence of a disulphide bond, this motion in $L_{5,6+7}$ propagates, resulting in the eventual unfolding of the β5 and β6 + 7 strands in the free monomer. The disulphide bond (C137), $L_{5,6+7}$ (D129, E130), and H124 are indicated

Our CPMG RD data inform on a specific mechanism for monomer release. The loop $L_{5,6+7}$ located at the dimer interface of HSP27 undergoes redox-independent motions on the ms timescale, and exists in a minor conformation that is unfolded (Fig. 4, Supplementary Table 1). When the disulphide bond is present, the local unfolding does not propagate further. However, in the reduced form and C137S variant, the disordering process extends, on the same timescale, into both the end of the β5 and beginning of the β6 + 7 strands, effectively destabilising the interface and facilitating monomer release (Fig. 7). Given the conservation of sHSP residues in $L_{5,6+7}$ (Fig. 7, Supplementary Fig. 10) and the previously observed ms motions in this region of cABC[68], transient unfolding of $L_{5,6+7}$ and the adjacent strands upon monomerisation is likely a common property of mammalian sHSPs.

We analysed the positions of 28 mutations in HSP27 that cause either CMT or dHMN (Fig. 7a), including the 17 missense mutations that reside in the ACD[55]. Our structural and dynamical analysis of the cHSP27 monomer reveals that 14 of the 17 mutations in the ACD are in or near the disordered region within $L_{5,6+7}$ and the β5 and β6 + 7 strands. As previously noted[28], a number of these mutations cluster to the ACD dimer interface. Our NMR data further reveal that the mutations that occur in regions beyond the dimer interface, predominantly fall in regions that are highly disordered in the monomer, suggesting that the behaviour of the monomer is important for understanding the molecular bases of CMT disease and dHMN[69]. Such significance could manifest perhaps in terms of altered activity, abundance, or through causing uncontrolled self-aggregation (Fig. 3).

While certain mutations decrease chaperone activity, some disease-related HSP27 variants that are more monomeric (e.g., R127W, S135F) exhibit significantly elevated chaperone activity both in vitro and in vivo[29,30], with increased affinity for substrate proteins (e.g., R140 mutation)[70]. Conversely, disease-related mutants that did not impact monomerisation have shown either no change (T151I) or a decrease (P182S/L) in activity[29,30,71]. Recent in vitro work on CMT-related HSP27 variants has demonstrated that ACD mutations can both enhance the dissociation of oligomers into smaller species and increase the overall size of the oligomers[72–74]. As the concentration of HSP27 in healthy human cells under basal conditions should be in the high nM to low μM range[75], free monomers would therefore be readily populated. These observations suggest that the strength of the monomer/dimer interface, and the monomer/dimer/oligomer equilibria are important factors for understanding neuropathies associated with variants of HSP27.

In light of our findings, we hypothesise that partial unfolding of sHSP ACDs upon monomer release may be a general feature of this class of chaperone. A recent study of cABC showed that the chemical shift changes upon monomer formation are larger for residues located at the dimer interface[68]. We analysed the data and found a strong correlation between $^{15}N$ chemical shift changes in cABC upon monomer formation with those expected for the formation of a random coil (Supplementary Fig. 10). More generally, the dimeric structure and sequence composition of residues at the ACD dimer interface are highly conserved in the mammalian sHSPs HSP27, ABC, AAC, and HSP20 (Fig. 7e, Supplementary Fig. 10). These results suggest that partial unfolding of monomers upon dissociation is a common property of human sHSPs and that the dimeric building block of sHSP oligomers is assembled first through partly unfolded monomers. Odd-numbered sHSP oligomers[21], as encountered in both human sHSPs ABC and HSP27, will have at least one monomer without a complete dimer interface, indicating that the unstructured monomer can also exist within larger oligomers. Even-numbered oligomers can have two or more monomers without a complete

dimer interface. Interestingly, for the related ABC, we observed that the dimeric form was more chaperone-active than the monomer, particularly for an amyloidogenic substrate[35]. These differences between the two sHSPs could reflect their contrasting substrate profiles[23], or multiple binding modes[76].

In the context of the isolated ACD, our data suggest that increased disorder in the HSP27 monomer renders it a more potent chaperone in vitro. In addition to the partially unfolded ACD, full-length HSP27 contains a disordered 80-residue NTD[77] and a highly flexible CTR comprising 28 residues[78,79]. More generally, these disordered regions are important for stabilising the oligomeric forms[80] and also contribute to chaperone activity. The inherent plasticity in these disordered regions, combined with the disorder in the monomeric ACD would, in principle, allow for the sampling of a wide range of conformational space and thereby facilitate its ability to interact with a diverse set of misfolded target proteins. As the monomer is itself prone to aggregation (Fig. 3d), we speculate that the aggregation-prone contacts in HSP27 are largely responsible for detecting misfolded proteins. In the context of the cell, it is would seem undesirable to have high concentrations of aggregation-prone monomers, making it advantageous to store them in oligomers that are sensitive to environmental conditions. By holding monomers in this 'storage' form, the population of the active but unstable monomeric form is kept both transiently low and highly available[28,41].

If the reduced, monomeric form of HSP27 is more potent than its oxidised counterpart, then how can the chaperone be a redox sensor inside living cells? We speculate that the monomeric form of HSP27 acts as a homoeostatic chaperone under basal conditions, in which this highly chaperone-active state can efficiently recognise misfolded proteins and prevent their aggregation (Figs. 1b and 2). In addition, the lone cysteine residue in HSP27 has been shown to be essential for its anti-apoptotic interaction with cytochrome C[81], suggesting that redox changes may play a role in the ability of HSP27 to influence cell death pathways.

In conclusion, our analysis combining CPMG RD and high-pressure NMR with chaperone and aggregation assays provides a structural characterisation of the sparsely populated and experimentally elusive monomeric form of HSP27. We demonstrate through redox- and concentration-dependent aggregation assays that the monomeric state is particularly active in vitro. Part of the region that forms the rigid interface in the dimer unfolds and becomes highly dynamic in the monomer, with the additional structural plasticity in the monomer rendering it a more effective chaperone. The monomer itself is more aggregation-prone, whereby off-pathway self-assembly culminates in uncontrolled aggregation. A survey of extant sHSP structures suggests that the exposure of the disordered, yet functional dimer interface, either in the context of oligomers or free in solution, is widespread across all sHSPs in all kingdoms of life.

## Methods

**Protein expression and purification**. All media for growth of E. coli containing pET(HSP27) plasmids contained 100 μg mL$^{-1}$ of Amp. Glycerol stocks of Amp-resistant (Amp$^R$) pET(HSP27) plasmids were used to inoculate 5 mL cultures for expression of full-length HSP27. Following growth at 37 °C for 6 h, the 5 mL cultures were transferred to 100 mL (LB medium) or 50 mL (M9 minimal medium) cultures and grown overnight at 37 °C, which were then used to inoculate 1 L (LB medium) or 500 mL (M9 minimal medium) cultures. When the optical density at 600 nm (OD$_{600}$) of these cultures reached between 0.6 and 0.8 units, IPTG was added to a final concentration of 100 μgmL$^{-1}$ and protein expression ensued for 3 h at 37 °C. Cells were pelleted and frozen at −80 °C until use. For preparation of uniformly-$^{13}C$, $^{15}N$-labelled ([U-$^{13}C$,$^{15}N$]-) HSP27, the M9 minimal medium contained 2 g L$^{-1}$ of [U-$^{13}C$]-glucose and 1 g L$^{-1}$ of $^{15}NH_4Cl$.

HSP27 was purified[82] using anion exchange chromatography (AEX) with HiTrapQ HP columns (GE Healthcare). The column was equilibrated in 20 mM Tris-HCl, 1 mM EDTA, pH 7 (AEX buffer A), after which the lysate was applied.

The column was washed with 5% of IEX buffer B (AEX buffer A with 1 M NaCl), and a linear gradient of 5–40% AEX Buffer B followed. HSP27 eluted around 15% AEX Buffer B (150 mM NaCl), after which all HSP27-containing fractions were pooled and concentrated for size exclusion chromatography (SEC). A Superdex S200 26/60 column (GE Healthcare) equilibrated in 20 mM Tris-HCl, 150 mM NaCl, 2 mM EDTA at pH 7 was used to separate HSP27 from the remaining contaminants. HSP27 eluted off the Superdex S200 26/60 near 200 mL, or 0.63 column volumes. After pooling HSP27-containing fractions after SEC, the samples were further purified with a second AEX step using a HiTrap Capto Q ImpRes column (GE Healthcare), with AEX Buffers A and B as described above. Purified HSP27 was then buffer exchanged using 10 K MWCO Amicon spin filters into 30 mM NaH$_2$PO$_4$, 2 mM EDTA, 2 mM NaN$_3$, pH 7.0 with 6% D$_2$O (NMR buffer), with an additional 100 mM NaCl for the oxidised state. To study the reduced state, 5 mM BME was added (Fig. 1, Supplementary Fig. 1). Oxidation was confirmed by both SDS-PAGE and native MS under non-reducing conditions.

For expression of cHSP27, DNA encoding for residues 84–171 of HSP27 (cHSP27) was inserted into kanamycin-resistant (Kan$^R$) pET28-b plasmids, which contained an N-terminal hexahistidine (His$_6$)-tag followed by a TEV protease recognition site[35]. The residual glycine that remains after TEV protease cleavage corresponds to G84 in the HSP27 amino acid sequence. All media for growth of cells containing pET28-b(cHSP27) plasmids contained 30 μgmL$^{-1}$ of Kan. cHSP27 cultures were grown as described above. Upon inoculation of the 1 L (LB medium) or 500 mL (M9 minimal medium) cultures, A600 was allowed to reach between 0.6 and 0.8 units, upon which IPTG was added to a final concentration of 100μgmL$^{-1}$ and protein expression continued for 3 h at 37 °C. Cells were pelleted and frozen at −80 °C until use. For preparation of isotopically labelled cHSP27, cells were grown in H$_2$O-based M9 minimal medium as above, supplemented with 4 g L$^{-1}$ of unlabelled glucose, 2 g L$^{-1}$ of [U-$^{13}$C]-glucose, 1 g L$^{-1}$ of $^{15}$NH$_4$Cl, or grown in 99.9% D$_2$O, respectively, for [U-$^{15}$N]-, [U-$^{13}$C,$^{15}$N]-, or fractionally deuterated (frac$^2$H), [U-$^{15}$N]-cHSP27.

cHSP27 was purified[35] with initial separation on a Ni$^{2+}$ column that was equilibrated in 20 mM Tris, 150 mM NaCl, 30 mM imidazole, pH 8.0 (IMAC Buffer A). Elution of bound protein from the column was achieved using IMAC Buffer A with 300 mM imidazole, and cHSP27-containing fractions were pooled and incubated with His$_6$-tagged TEV protease in IMAC Buffer A at room temperature overnight with 5 mM β-mercaptoethanol (BME) to cleave the His$_6$-tag from cHSP27. Cleaved cHSP27 was then separated from His$_6$-tagged TEV protease, the cleaved His$_6$-tag, and other contaminants using reverse Ni$^{2+}$ affinity separation and then subjected to SEC on a Superdex S75 26/60 column (GE Healthcare), eluting at 170–180 mL, or 0.53–0.56 column volumes. The final SEC buffer contained 20 mM Tris-HCl, 150 mM NaCl, and 5 mM BME at pH 7.0. Using 3 K MWCO Amicon spin filters, protein samples were buffer exchanged into and concentrated into NMR buffer for the oxidised state at pH 7 unless otherwise specified. 5 mM BME was added when studying the reduced state (Figs 3–7, Supplementary Fig. 3–8).

Primers were designed to encode the single-point mutation C137S in cHSP27, and this plasmid was used to generate the double variant H124K/C137S via site-directed mutagenesis (Supplementary Table 6). Site-directed mutagenesis was performed using the QuikChange II Mutagenesis Kit (Agilent), and the correct mutations were verified by DNA sequencing. Protein expression and purification was carried out as above. The final yield of the double mutant, cHSP27(H124K/C137S) (~10 mgL$^{-1}$ of E. coli) was significantly lower than either WT cHSP27 or C137S (30–40 mgL$^{-1}$ of E. coli).

**Chaperone activity assays.** All chaperone activity assays were completed using a 96-well plate and a FLUOstar Omega Microplate Reader or Tecan Infinite M200 PRO plate reader. Chaperone activity was defined as one minus the ratio of signal observed in the presence of chaperone to that in the absence of chaperone, e.g., values of 1 and 0, respectively indicate complete protection against aggregation and incapability to prevent aggregation. To avoid selecting a single time point or averaging a few time points for the analysis of chaperone activity, we calculated the integral under the observed curves of absorbance or fluorescence versus time. The integral under the curve was determined for each of the replicate wells for a given substrate and substrate/chaperone combination, and the reported chaperone activity values in Figs. 1–3 reflect the average and one standard deviation. The y axes for plots of aggregation versus time were normalised on a scale of 0–1, with 1 set to the maximum value obtained for substrate alone. The x axes were likewise normalised on a scale of 0–1 by dividing time by a factor $T_f$, which varied for each substrate

Porcine heart CS (Sigma-Aldrich) was buffer exchanged into NMR buffer supplemented with 100 mM NaCl. The aggregation of CS is initiated at high temperatures and does not require the addition of a chemical denaturant (e.g., reducing agents), which enables a direct comparison of the chaperone activity of reduced and oxidised HSP27. However, CS aggregates faster under reducing conditions (Supplementary Fig. 2), complicating interpretation. The aggregation of 10 μM CS was monitored at 43 °C by following light scattering at 340 nm as a function of time. HSP27 was added to a final concentration of 0.5 μM, a concentration low enough for a substantial concentration of monomers to be expected[32]. The assays were conducted in the absence or presence of 5 mM BME. Assays were completed in duplicate and the mean ±1 standard deviation is reported. The light scattering traces in the presence of reduced and oxidised

HSP27 shown in Fig. 1 are normalised to the respective aggregation of reduced and oxidised CS alone, and plotted on the same axis.

Porcine heart MDH and rabbit muscle GAPDH (Sigma-Aldrich) were respectively buffer exchanged into or dissolved in NMR buffer, and aggregation assays were conducted at 40 °C at 0.2 μM (MDH) or 2 μM (GAPDH) final concentration. Aggregation was monitored by measuring light scattering at 340 nm as a function of time, in the presence or absence of 5 mM BME. The redox state of MDH and GAPDH did not affect aggregation (Supplementary Fig. 2), enabling a direct comparison between the chaperone activity of reduced and oxidised HSP27, which was added in final concentrations of 0.5, 2, or 20 μM. Assays were conducted in triplicate and the mean ±1 standard deviation is reported.

Human αS was recombinantly expressed and purified using AEX chromatography followed by gel filtration chromatography[39]. All samples were prepared in PBS with a final concentration of 200 αS μM, 4 μM ThT, and 10 μM αS fibril seeds in the presence or absence of 1 mM DTT. The αS fibril seeds were prepared through ThT-based fibrillation assays and confirmed by EM. Amyloid fibril formation in the seeded assays was determined by ThT fluorescence with excitation at 440 nm and emission at 485 nm at 37 °C over one week. All fluorescence experiments were performed using a Corning 3881 96-well plate.

Calcium-depleted bovine α-Lac and human insulin (Sigma-Aldrich) were dissolved in NMR buffer, and aggregation was initiated by the addition of 2 mM DTT followed by incubation at high temperatures. The aggregation of αLac (300 μM) at 37 °C and insulin (80 μM) at 40 °C was initiated upon the addition of 1 mM DTT, and aggregation was monitored by measuring light scattering at 340 nm as a function of time. cHSP27 variants were added to a final concentration of 70 μM (αLac) or 40 μM (insulin), a concentration where C137S and reduced cHSP27 are predominantly dimers and H124K/C137S a monomer (Fig. 7, Supplementary Fig. 8). Assays were conducted in triplicate and the mean ± one standard deviation is reported.

The auto-aggregation of H124K/C137S was monitored by following the absorbance at 340 nm at 37 °C in NMR buffer. The total volume was 100 μL and the protein concentration was either 200 or 800 μM. For comparison, C137S was prepared at 800 μM and subjected to identical treatment. Notably, the sequence-based aggregation propensity predictors Tango[83] and Zyggregator[84] do not predict any notable change owing to the choice of mutation. All concentrations had either three or six replicates, and the mean ± SD is reported in Fig. 3 and Supplementary Fig. 4.

**Native MS.** For native MS data acquisition, nanoelectrospray ionisation (nESI) experiments were executed according to published protocols using instrument settings optimised to transmit intact protein complexes[85]. A 25 μM (monomer concentration) sample of HSP27, which was purified in the absence of reducing agent, was prepared in 200 mM ammonium acetate (pH 6.9) in the presence and absence of 250 μM DTT (Fig. 1, Supplementary Fig. 1). cHSP27 samples (5 μM) were prepared in the same buffer with and without 250 μM DTT (Fig. 3). Data were collected on Q-ToF 2 (Micromass/Waters Corp.) and Synapt G1 (Waters Corp.) instruments.

**Thermal melts.** Thermal melts were performed with a nanoDSF instrument (NanoTemper), which was used to monitor the intrinsic fluorescence of cHSP27 and C137S as a function of temperature. Capillaries contained ~10 μL of cHSP27 that had been prepared in NMR buffer without (oxidised) or with 5 mM BME (reduced). C137S was prepared in NMR buffer without BME. The concentrations of cHSP27 and C137S were 1000, 100, 10, and 1 μM. The initial temperature was 20 °C and was set to increase by 1 °C per minute. Fluorescence readings were recorded at 330 and 350 nm, and the melting temperature (T$_m$) recorded (Supplementary Fig. 7).

**CD spectroscopy.** Samples for CD spectroscopy were prepared at 20 μM in NMR buffer and placed into cuvettes with a 1 cm path length. Data were recorded using a Jasco Model J720 CD spectrophotometer at wavelengths between 200 and 260 nm. The fluorescent probe bis-ANS (Sigma Aldrich) was dissolved in NMR buffer and added to a final concentration of 10 μM to wells containing a final volume of 100 μL of 40 μM C137S or H124K/C137S in NMR buffer. Under these conditions, C137S is primarily a dimer and H124K/C137S is primarily a monomer. Fluorescence at 500 nm was recorded following excitation at 350 nm.

**NMR spectroscopy.** All NMR spectroscopy experiments for resonance assignments of cHSP27 and HSP27 at ambient pressure were recorded on a 14.1 T Varian Inova spectrometer equipped with a 5 mm z-axis gradient triple resonance room temperature probe. 2D $^1$H-$^{15}$N sensitivity-enhanced HSQC spectra[86] at 14.1 T were typically acquired with $^1$H ($^{15}$N) 513 (64) complex points, spectral widths of 8012 Hz (1800 Hz), maximum acquisition times of 64 ms (35.6 ms), an inter-scan delay of 1 s, and four scans per FID for a total acquisition time of 19 min. [U-$^{13}$C,$^{15}$N]-cHSP27 was prepared at a final concentration of 1 mM in NMR buffer, and [U-$^{13}$C,$^{15}$N]-HSP27 was prepared at a final concentration of 2 mM in NMR buffer supplemented with 100 mM NaCl. Standard assignment experiments[87] were collected, specifically:

HNCO: 766/50/25 complex points and 85.1/39.8/23.6 ms maximum acquisition times for H/C/N dimensions respectively, with 8 scans per FID for a total acquisition time of 12.76 h.

HN(CA)CO: 1024/40/30 complex points and 85.1/10.6/13.3 ms maximum acquisition times for H/C/N, with 16 scans per FID for a total acquisition time of 23.66 h.

HNCA: 766/40/30 complex points and 85.1/14.8/28.4 ms maximum acquisition times for H/C/N, with 16 scans per transient for a total acquisition time of 24.07 h.

C(CO)NH: 577/40/20 complex points and 64/4.4/18.9 ms maximum acquisition times for H/C/N, with 64 scans per FID for a total acquisition time of 61.86 h.

HN(CO)CA: 1024/50/20 complex points sampled at 25% sparsity (H/C/N), 85.1/18.5/18.9 ms maximum acquisition times for H/C/N, with 32 scans per FID for a total acquisition time 8.8 h.

When NUS was employed, an exponentially weighted sampling scheme was employed in the indirect dimensions and time-domain data were reconstructed with MddNMR[88]. All 3D NMR spectra at ambient pressure were acquired at 25 °C, processed with NMRPipe[89], and visualised with Sparky[90]. The resultant $^1$H$^N$, $^{15}$N, $^{13}$CO, $^{13}$Cα, and $^{13}$Cβ chemical shifts were analysed with TALOS-N[91] and RCI[92] to respectively estimate the secondary structure and N−H order parameters (Supplementary Fig. 3).

To indirectly probe hydrogen bonds in oxidised cHSP27, $^1$H-$^{15}$N HSQC spectra of [U-$^{15}$N]-oxidised cHSP27 were recorded at 293, 295, 298, and 303 K, and the $^1$H chemical shift temperature coefficients (dω$^1$H/dT) were calculated (Supplementary Fig. 2). dω$^1$H/dT values that are more negative than –4.6 ppbK$^{-1}$ are more likely to be solvent exposed and hydrogen bonded to water[93]. Residues with temperature coefficients more positive than –4.6 ppbK$^{-1}$ are more likely to be involved in intra- or inter-protein hydrogen bonds. However, it should be noted that residues that are near aromatic rings can yield false positives with values less than –4.6 ppbK$^{-1}$ [93].

To provide an independent NMR dataset that also indirectly probes hydrogen bonds, we buffer exchanged a sample of 1 mM [U-$^{15}$N]-oxidised cHSP27 into 99.9% D$_2$O and recorded a 2D $^1$H-$^{15}$N HSQC spectrum. The dead time was ~40 min for the entire process and the sample was kept at 4 °C during this time. Intra- and inter-cHSP27 hydrogen bonds involving amide protons were assessed by the presence or absence of signals in the D$_2$O $^1$H-$^{15}$N HSQC spectrum (Supplementary Fig. 3).

Resonance assignments for monomeric C137S were obtained on a 14.1 T Bruker Avance-III spectrometer equipped with a cryogenic probe. A [U-$^2$H,$^{13}$C,$^{15}$N]-labelled sample at pH 4.1, 2 mM sodium phosphate, 25 °C was prepared at 100 μM. 3D HNCA and HNCO spectra were acquired under these conditions. The HNCO spectrum was acquired with 10% NUS, an inter-scan delay of 1.5 s, 1024/170/140 complex points (H/C/N) with maximum acquisition times of 104/70/85.5 ms, and 4 scans per FID for a total acquisition time of 16 h. For the HNCA spectrum, 1024/115/84 complex points were acquired for H/C/N with maximum acquisition times of 104/20/58 ms, with 8 scans per FID and an inter-scan delay of 1.4 s for a total acquisition time of 81 h. A similar set of experiments was recorded on [$^2$H,$^{13}$C,$^{15}$N]-labelled C137S dimer at pH 7 in order to compare $^{13}$C chemical shifts. The NUS spectra were reconstructed with SMILE[94].

For measurement of spin relaxation experiments, standard pulse sequences to measure $^{15}$N heteronuclear nuclear Overhauser enhancements (hetNOE), longitudinal ($T_1$), and transverse ($T_2$) relaxation times were employed[95], using 14.1 and 11.7 T Varian Inova spectrometers equipped with 5 mm z-axis gradient triple resonance room temperature probes. All experiments at 14.1 T were recorded with 576 (32) complex points, 8992 Hz (1800 Hz) sweep widths, and 64 ms (18 ms) acquisition times, for $^1$H ($^{15}$N). All experiments at 11.7 T were recorded with 512 (32) complex points, 8000 Hz (1519 Hz) sweep widths, and 64 ms (21 ms) acquisition times, for $^1$H ($^{15}$N). $T_1$ and $T_2$ values are reported as their respective inverses, i.e., rates ($R_1$ and $R_2$), for convenience. The spectrometer temperature was calibrated with d$_4$-methanol. On oxidised cHSP27, we recorded $^{15}$N $R_1$ (11.7, 14.1 T), $R_2$ (11.7, 14.1 T), and hetNOE (14.1 T only) values at 25 °C on a 1 mM sample of [U-$^{13}$C,$^{15}$N]-labelled sample in NMR buffer (Supplementary Fig. 4). Upon completion of these measurements, 5 mM β-mercaptoethanol (BME) was added to the NMR tube, and $^{15}$N $R_1$, $R_2$, and hetNOE values were recorded on reduced cHSP27 at 600 MHz only (Supplementary Fig. 4). Similarly, $^{15}$N $R_1$, $R_2$, and hetNOE experiments at 14.1 T were recorded on oxidised [U-$^{13}$C,$^{15}$N]-labelled full-length HSP27, after which 5 mM BME was added to obtain relaxation measurements on the reduced and oxidised species (Supplementary Fig. 1). For data acquired exclusively at 14.1 T, reduced spectral density functions were calculated as described[96]. For data acquired at multiple magnetic field strengths (oxidised cHSP27), a model-free analysis was conducted (see below). All data sets mentioned here were processed with NMRPipe[89], visualised with Sparky[90], and peak shapes were fit and analysed with FuDA[97]. hetNOE experiments on the C137S monomer and dimer were performed on a 14.1 T Bruker Avance-III spectrometer equipped with a cryogenically cooled probe using a published pulse sequence[98].

$^{15}$N relaxation data from oxidised cHSP27 at 11.7 T (500 MHz) and 14.1 T (600 MHz) were analysed using Lipari–Szabo models[99,100]. Prior to fitting the relaxation data with various Lipari–Szabo models[99,100], we first determined the rotational diffusion behaviour of cHSP27. Residues with evidence of enhanced ps–ns or μs–ms dynamics, as respectively determined by hetNOE values < 0.65 and $R_2$ values > 1 SD from the mean, were excluded from the diffusion tensor analysis

according to Tjandra et al.[101]. After removing dynamic residues, the remaining $^{15}$N $R_2/R_1$ ratios were fitted to isotropic ($D_{xx} = D_{yy} = D_{zz}$), axially symmetric ($D_{xx} = D_{yy} \neq D_{zz}$), and anisotropic ($D_{xx} \neq D_{yy} \neq D_{zz}$) diffusion tensors, which were statistically validated by comparing values of $\chi^2_{red}$ and by using F-tests. This analysis indicated that axial symmetry provided the most appropriate diffusion model for cHSP27, which was used to fit $^{15}$N $R_2/R_1$ ratios with ModelFree4.15[102] to determine a global correlation time ($\tau_c$) of 13.65 ns, which was consistent with a ~20 kDa dimer that exhibits slower diffusion of the axis parallel to the principal component of the diffusion tensor ($D_{||}/D_{\perp} = 1.52$). For this analysis, a crystal structure of cHSP27 (PDB 4mjh[35]) was rotated into the frame of the diagonalised diffusion tensor. Values for $\tau_c$ and $D_{||}/D_{\perp}$ were fixed in the subsequent model-free analysis, in which the N–H bond length was set to 1.02 Å and the $^{15}$N CSA was set to −160 ppm. Using the five aforementioned values of $^{15}$N relaxation rates from two magnetic fields, we fit these data using ModelFree4.15[102] to four models: model 1 (optimised $S^2$), model 2 ($S^2$ and $\tau_e$), model 3 ($S^2$ and $R_{ex}$), and model 4 ($S^2$, $\tau_e$, and $R_{ex}$). No further benefit was obtained when $S^2_f$ was included in the fitting, and thus was not required for this analysis.

$^{15}$N CPMG relaxation dispersion (RD) experiments were recorded on 11.7 and 14.1 T NMR spectrometers at 25, 30, and 35 °C using standard pulse sequences[103]. Each data set was recorded with $^1$H ($^{15}$N) 615 (35) complex points (512/29 at 11.7 T), sweep widths of 9615 Hz (1800 Hz) (8000 Hz/1519 Hz), acquisitions times of 64 ms (19 ms) (64 ms/19 ms), and a relaxation delay of 3 s. Each experiment was recorded as a pseudo-3D spectrum with the third dimension encoded by the variable delay between pulses in the CPMG pulse train ($\tau_{CPMG} = 4\nu_{CPMG}^{-1}$). For these measurements, frac$^2$H, [U-$^{15}$N]-labelled samples of cHSP27 (1 mM) or C137S (1 mM and 0.3 mM) were prepared. The experiments contained a fixed constant time of 39 ms for the CPMG period and 20 values of $\nu_{CPMG}$ ranging from 54 Hz to 950 Hz. Four scans (1 mM) or eight scans (0.3 mM) per FID were recorded for a total acquisition time of ~10 h (1 mM) or ~20 h (0.3 mM). Peak shapes were fit with FuDA[97] to extract peak intensities, which were then converted into $R_{2,eff}$ values using the following relation:

$$R_{2,eff}(\nu_{CPMG}) = \frac{-\ln\left(\frac{I_{\nu_{CPMG}}}{I_0}\right)}{T_{relax}} \qquad (1)$$

where $I_{\nu_{CPMG}}$ is the intensity of a peak at $\nu_{CPMG}$, $T_{relax}$ is the constant relaxation delay of 39 ms that was absent in the reference spectrum, and $I_0$ is the intensity of a peak in the reference spectrum. Two duplicate $\nu_{CPMG}$ points were recorded in each dispersion data set for error analysis, and uncertainties in $R_{2,eff}$ were calculated using the standard deviation of peak intensities from such duplicate measurements. From plots of $R_{2,eff}$ as a function of $\nu_{CPMG}$, $R_{ex}$ was estimated by taking the difference of $R_2(54 Hz)$ and $R_2(950 Hz)$. The program CATIA[97] was used for analysis (Supplementary Table 1), which accounts for all relevant spin physics including imperfect 180º pulses and differential relaxation between spin states[97], factors that are not accounted for when using closed form analytical solutions[104]. For each combination of concentration and temperature, data were analysed over multiple field strengths for all residues where $R_{ex} > 2 s^{-1}$. All residues were assumed to experience a 'two-state' equilibrium between a majorly populated 'ground' state and a sparsely populated 'excited' conformational state, and so the same interconversion rates were applied to all residues during analysis. Uncertainties in fitted parameters were estimated by a boot-strapping procedure where synthetic datasets were created by sampling residues with replacement, re-analysing the new dataset, and storing the result. The distribution of parameters achieved from 1,000 such operations provided a measure of experimental uncertainty.

Twenty-five residues with $R_{ex} > 2 s^{-1}$ at 14.1 T were selected for further analysis (Supplementary Table 2), and their $^{15}$N CPMG RD data at 14.1 and 11.7 T were globally fit to a model of two-site chemical exchange using the program CATIA[97]. This analysis enabled extraction of $k_{ex}$, $p_E$, and $\Delta\omega$ values (Supplementary Table 1, Supplementary Table 2). The global fit from all 26 residues yielded a reduced $\chi^2$ of 1.46. These residues comprised V101, D107, T110, D115, G116, V117, T121, G122, H124, E125, E126, R127, D129, E130, G132, Y133, I134, S135, R136, S137, F138, T139, R140, K141, and T143. The resultant $^{15}$N $|\Delta\omega|_{CPMG}$ values from C137S were compared to $^{15}$N chemical shift changes between the folded C137S dimer and random coil conformations ($^{15}$N $|\Delta\omega|_{RC}$). The four random coil $^{15}$N chemical shift databases within the neighbour-corrected intrinsically disordered library (ncIDP[20]) were employed, and the average $^{15}$N random coil chemical shift and standard deviation was used as the $^{15}$N $|\Delta\omega|_{RC}$, which was then correlated with the measured values of $^{15}$N $|\Delta\omega|_{CPMG}$ (Supplementary Fig. 8).

For oxidised cHSP27, residues D129, E130, R136, C137 and F138 showed significant variation in $R_{2,eff}$ with $\nu_{CPMG}$ (Supplementary Table 3, Supplementary Table 4) ($R_{ex} > 2 s^{-1}$). Analysis assuming each residue had independent exchange rates revealed that D129 and E130 clustered with a $k_{ex}$ ~1250 s$^{-1}$ (Supplementary Table 3), whereas R136, C137 and F138 formed a cluster with $k_{ex} > 3000 s^{-1}$ (Supplementary Table 4). This distinction was apparent from the raw data (Supplementary Fig. 6) as CPMG RD curves from the latter group showed variation in $R_{2eff}$ with $\nu_{CPMG}$ at high pulse frequencies, whereas those from the first group were effectively in the fast exchange limit at much lower CPMG frequencies. The reduced chi squared value, $\chi^2_{red}$ of the first group was 1.01, and that of the second group was 1.02.

For reduced cHSP27, CPMG RD data were acquired at 14.1 T exclusively. Residues with $R_{ex} > 2\,s^{-1}$ were globally fit to a model of two-site chemical exchange, assuming a monomer-dimer exchange event, as above. Twenty-three residues were included in the fit (Supplementary Fig. 6, Supplementary Table 5), comprising V101, D107, T110, D115, G116, V118, T121, G122, K123, H124, E125, E126, R127, D129, E130, Y133, I134, S135, F138, T139, R140, K141, and T143. Residues R136 and C137 were not analysed because these resonances were too broad to provide reliable measurements. The $\chi^2_{red}$ value from the global fit at one magnetic field strength was 1.45, with $k_{ex}$ of ~1500 $s^{-1}$ and $p_E$ of ~2%.

The CPMG data were analysed according to a two-state equilibrium scheme where $G \rightleftharpoons E$ with the forward reaction rate termed $k_{eg}$ and the reverse $k_{ge}$. As described in the text above, $k_{eg}$ was found to be concentration dependent, allowing identification of the minor state E to be monomeric, suggesting the equilibrium has the form $A_2 \rightleftharpoons 2A$ with the forward rate termed $k_{off}$ and the backward rate $k_{on}$. The two-state rate constants derived from CPMG measurements, $k_{eg}$ and $k_{ge}$, were converted to $K_d$, $k_{on}$ and $k_{off}$ measurements as described in Supplementary Table 1. The $K_d$ values obtained from CPMG analysis from C137S using data acquired at 1 mM and 0.3 mM were highly similar (Supplementary Fig. 6), supporting the identification of the equilibrium to be monomer/dimer exchange.

For extraction of thermodynamic parameters of the exchange event between the C137S dimer and monomers, $^{15}N$ CPMG RD data were recorded at multiple temperatures. Namely, we recorded data on a 1 mM sample of $^2H$, [$U$-$^{15}N$]-C137S at 25, 30, and 35 °C at both 11.7 T and 14.1 T and analysed globally, assuming the forward and backward rates can be described by the Eyring equation, as can the equilibrium constant, as indicated above in Supplementary Table 1 and Eqs. 7–9 [105]. Justifying this, locally obtained values of $k_{eg}$, $k_{ge}$, and $K_{eq}$ vary with temperature in a manner consistent with these equations (Supplementary Fig. 6). An additional dataset was recorded at 0.3 mM at 25 °C at both 11.7 and 14.1 T. For each combination of concentration and temperature, the data were globally analysed in terms of a two-state model to extract thermodynamic and kinetic parameters. Data were analysed according to the following equilibrium scheme between a 'ground' ($G$) and an energetically 'excited' ($E$) conformational state, $G \rightleftharpoons E$ where the interconversion rate is:

$$k_{ex} = k_{eg} + k_{ge} \tag{2}$$

with $k_{eg}$ and $k_{ge}$ describing the reaction rates for the conversion of state $G$ to state $E$ (forward rate) and vice versa. The fractional population of the excited state is:

$$p_E = \frac{k_{ge}}{k_{ex}} \tag{3}$$

By analysing the CPMG RD data at multiple concentrations, $k_{eg}$ and $p_E$ were found to be concentration-dependent whereas $k_{ge}$ was not. This led to the hypothesis that the equilibrium scheme was a monomer/dimer association of the form $A_2 \rightleftharpoons 2A$, with the forward rate termed $k_{on}$ and the reverse rate termed $k_{off}$, where the excited state corresponds to the monomer. Within this scheme, the rates are related as follows:

$$k_{eg} = 2k_{on}[A] \tag{4}$$

$$k_{ge} = k_{off} \tag{5}$$

The dissociation equilibrium constant ($K_d$) is:

$$K_d = \frac{[A]^2}{[A_2]} = \frac{2p_E^2[Tot]}{1 - p_E} \tag{6}$$

where [$Tot$] is the total monomer concentration. Through analysis of the CPMG data, $p_E$ and $k_{ex}$ values are obtained, which can then be related to the dimer specific equilibrium constants. The specific residues used for C137S analysis, and chemical shift differences obtained from the CPMG analysis are tabulated for the 25 °C, 1 mM dataset (Supplementary Table 2).

From Eyring theory, chemical rates can be related to enthalpy and entropy changes of activation, the changes associated with moving from the stable state to the transition state.

$$k_{eg} = ATe^{-\left(\frac{\Delta H_{eg} - T\Delta S_{eg}}{RT}\right)} \tag{7}$$

$$k_{ge} = ATe^{-\left(\frac{\Delta H_{ge} - T\Delta S_{ge}}{RT}\right)} \tag{8}$$

with $A$ the reaction frequency, set here to be 3000 $s^{-1}\,K^{-1}$, and $T$ is the thermodynamic temperature [105]. Similarly, the equilibrium constant is given by:

$$K_{eq} = \frac{[E]}{[G]} = e^{-\left(\frac{\Delta H - T\Delta S}{RT}\right)} \tag{9}$$

To monitor the pH-induced dissociation of C137S dimers, a 2D $^1H$-$^{15}N$ HSQC spectrum was recorded on a sample of [$U$-$^{15}N$]-C137S at 250 μM in NMR buffer at pH 7 at 25 °C. Separate samples were independently prepared in NMR buffer at pH

6.5, 6.0, and 5.0 and NMR spectra were recorded to assess the effect of pH on dimerisation. The well-resolved peak from G116 was used to calculate the dimerisation $K_d$ as a function of pH (Supplementary Fig. 6, inset). At pH 5, no dimer was observed, and thus the $K_d$ at pH 5 (~5 mM) is four orders of magnitude larger than that at pH 7 (0.5 μM). Below pH 6.5, the sample was highly unstable and white precipitate was evident by the end of the NMR experiments (20–40 min). Similar pH titrations were performed on oxidised cHSP27, with no evidence for population of a monomeric intermediate (Supplementary Fig. 7). From these pH titrations, the dimerisation dissociation constant ($K_d$) was determined. From peak intensities, the relative dimer and monomer ratios can be determined as a function of total protein concentration. These can be used to obtain an estimate for $K_d$:

$$K_d = \frac{[A]^2}{[A_2]} \tag{10}$$

From the detailed balance, the total concentration will be:

$$[Tot] = [A] + 2[A_2] \tag{11}$$

With a homo-dimer, the intensity of monomer ($S_A$) and dimer ($S_D$) can be written in terms of the mole fraction of monomer signal:

$$F = \frac{S_A}{S_A + S_D} \tag{12}$$

The raw signal intensity from the dimer will be proportional to the monomers in the dimer and so:

$$S_D = 2[A_2]k \tag{13}$$

whereas signal from the monomer will be proportional to free monomers:

$$S_A = [A]k \tag{14}$$

The mole fraction is equal to:

$$F = \frac{[A]}{[A] + 2[A_2]} \tag{15}$$

and thus the $K_d$ can be recast in terms of the total monomer concentration and the signal intensities from monomer and dimer:

$$K_d = \frac{2[Tot]F^2}{1 - F} \tag{16}$$

To monitor the pressure-induced dissociation of C137S dimers, a sample of 200 μM [$U$-$^{15}N$]-C137S was prepared in a baroresistant buffer[56] at pH 7 in order to prevent changes in pH with increasing pressure. The buffer contained a mixture of 100 mM Tris-HCl and 100 mM phosphate buffer, both at pH 7, and yields a negligible change in pH between 1 and 2500 bar. 2D $^1H$-$^{15}N$ HSQC spectra were recorded with $^1H$ ($^{15}N$) 1024 (100) complex points, 8417 Hz (2083 Hz) sweep widths, 121.7 ms (48 ms) acquisition times at 14.1 T at 25 °C as a function of hydrostatic pressure between 1 bar to 2500 bar. Manipulation of the hydrostatic pressure was carried out using a commercial ceramic high-pressure NMR cell and an automatic pump system (Daedalus Innovations, Philadelphia, PA). Peak intensities from non-overlapping signals arising from the dimer, monomer, and unfolded species were quantified at each pressure and fit to a model of three-state unfolding (Fig. 6, Supplementary Table 1). Similar pressure titrations were performed on oxidised cHSP27, with no evidence for population of a monomeric intermediate (Supplementary Fig. 7), although pressure-induced unfolding at high pressures was evident.

These HSQC spectra recorded as a function of pressure provide the relative NMR signals of the dimeric ($A_2$), monomeric ($A$) and unfolded monomeric states ($B$). As the system is in slow exchange, the data can be globally analysed using a three-state equilibrium model where $A_2 \rightleftharpoons 2A$ and $A \rightleftharpoons B$, with the equilibrium constant for dimer dissociation ($K_d$) equal to Eq. 6 and for unfolding ($K_U$) equal to:

$$K_U = \frac{[B]}{[A]} \tag{17}$$

The detailed balance of system means that the total concentration at any pressure is:

$$[Tot] = [A] + 2[A_2] + [B] \tag{18}$$

which can be recast into a function of only $A$ (monomer):

$$[Tot] = [A] + \frac{2[A]^2}{K_d} + K_U[A] \tag{19}$$

which can be solved to obtain:

$$[A] = \frac{-K_d}{4}(K_U + 1) + \frac{\sqrt{K_d^2(K_U + 1)^2 + 8[Tot]K_d}}{4} \quad (20)$$

For any specified $K_d$, $K_U$, and [Tot], [A] and hence $[A_2]$ and [B] can be determined. The free energy of each step is expected to vary with pressure as:

$$\frac{\partial \Delta G}{\partial P} = \Delta V \quad (21)$$

Assuming the three structures are incompressible, integrating the free energy yields:

$$\Delta G_{P_1} - \Delta G_{P_2} = \Delta V(P_2 - P_1) \quad (22)$$

This equation provides expressions for the variation of the equilibrium constants with pressure:

$$K_d = K_d^0 e^{-\left(\frac{\Delta V_{A_2 toA}(P - P_0)}{RT}\right)} \quad (23)$$

$$K_U = K_U^0 e^{-\left(\frac{\Delta V_{AtoC}(P - P_0)}{RT}\right)} \quad (24)$$

These equations allow for the mole fractions of $[A_2]$, [A] and [B] to be determined at a given $K_d^0$, $K_U^0$, $\Delta V_{A_2 toA}$ and $\Delta V_{AtoC}$ at a specified total concentration and temperature. Using this scheme, the NMR data that reports on these mole fractions was fitted to the model, to obtain $K_d^0$, $K_U^0$, $\Delta V_{A_2 toA}$ and $\Delta V_{AtoC}$ as fitting parameters.

**Random coil index**. To identify β strands in the C137S monomer and dimer, we utilised the software Random Coil Index[57] and Chemical Shift Index[58] with default settings. $^1H^N$, $^{13}CO$, $^{13}C\alpha$, and $^{15}N$ chemical shifts from the C137S dimer (pH 7) and monomer (pH 4.1) were used as input values. The deuterium isotope effect was corrected for using established values[106].

## Data availability
These data that support the findings of this study are available from the corresponding authors upon reasonable request.

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

## Acknowledgements

We thank Prof. Heath Ecroyd (University of Wollongong) for the WT HSP27 expression plasmid, Prof. Arthur Laganowsky (now Texas A&M University) for assistance in constructing the cHSP27 expression plasmid, Prof. Christina Redfield (University of Oxford) for usage of the CD spectrophotometer, Dr. Errin Johnson (University of Oxford) for assistance with electron microscopy and Dr. David Staunton (University of Oxford) for assistance with thermal denaturation experiments. T.R.A. acknowledges funding from the NIDDK, the NIH Oxford-Cambridge Scholars Program, and Pembroke College; J.L.P.B. thanks the Engineering and Physical Sciences Research Council (EP/J01835X/1) and Biotechnology and Biosciences Research Council (BB/J018082/1); and A.J.B. holds a David Phillips Fellowship from the Biotechnology and Biosciences Research Council (BB/J014346/1). This work was supported in part by the Intramural Research Program of the National Institutes of Diabetes and Digestive and Kidney Diseases and the Intramural Antiviral Target Program of the Office of the Director, NIH.

## Author contributions

Conceptualisation, T.R.A., A.B., J.L.P.B. and A.J.B. Investigation, T.R.A., J.R., H.Y.G., D.M.D., I.P., J.Y., J.L.P.B., A.B. and A.J.B. Writing – Original Draft, T.R.A., A.B., J.L.P.B. and A.J.B. Writing – Reviewing and Editing, T.R.A., J.R., H.Y.G., D.M.D., I.P., J.Y., A.B., J.L.P.B. and A.J.B.

## Additional information

**Competing interests:** The authors declare no competing interests.

