## [Peer Review File · Nature Communications]

Reviewers' comments:

Reviewer #1 (Remarks to the Author):

This is a very well written manuscript describing the use of a combination of CPMG relaxation dispersion and high-pressure NMR to study the dimer-monomer equilibrium of the conserved $\alpha\beta$ -crystallin domain (ACD) of Hsp27. The authors found that the ACD dimer exchanges with a sparsely populated partially disordered monomer that shows higher chaperone activity than the Hsp27 ACD dimers. The authors obtained structural information on this sparsely populated Hsp27 ACD monomer and showed that the monomer partially unfolds, most likely upon dissociation, and that the regions in the dimer interface are highly dynamic in the monomer. The results are well presented and overall the quality of the work is impressive. I am happy to recommend publication with some minor comments / suggestions.

1) The observation that the increased release of monomers into the solution is responsible for the higher chaperone activity of reduced Hsp27 is very interesting. It would be nice, however, to see a direct comparison of disaggregation suppressing activity between oxidized, reduced, and monomeric Hsp27 ADC using the same model aggregates, for example α -Lactalbumin, shown in figure 2C.

On a related note, can the authors speculate as to why the aggregation of citrate synthase is significantly enhanced in the presence of oxidized Hsp27?

2) The authors compare the chemical shift changes obtained through CPMG RD to dimer - random coil values of Hsp27 ADC loop 5,6+7 and β -sheet 5 and β -sheet 6+7. It would be beneficial to conduct a similar comparison between CPMG RD $\Delta\omega$ s and chemical shift differences between the wt dimer and Hsp27 H124K/C137S monomer.

3) While the structural data is very solid, the biological implications of this study are somewhat confusing. The oxidized state of Hsp27 was shown by several studies to be the active form of the chaperone and therefore to protect the cell from oxidative stress. How do the authors, then, resolve these previous findings with their own observation that the monomer, which is released mostly in the reduced form of Hsp27, is the active state?

On a similar note, the authors hypothesize that the local unfolding of the monomer is a general redox independent mechanism of monomer release. The authors further state that when the disulfide bond is present, the local unfolding does not result in monomer release from the dimer and that this mechanism might be conserved in other small heat shock proteins. This is an interesting idea, however, since all other heat shock family members, except Hsp27 (HspB1), don't form disulfides - then the amount of detectable monomers in other Hsps should be higher, since there is no disulfide to stabilize the dimer. Is there any experimental evidence showing higher monomer presence in other Hsps, or higher chaperone activity due to increased monomer ratio?

Minor comments / typos

1) Do the L5,6+7 loop residues in the reduced Hsp27 fit the same global monomer-dimer process as residues in β 5 and 6+7?

2) Main text, page 12 would mistakenly appear to reference figure 5bii instead of 5aii.

3) Main text figure legend for figure 5 - Fig5. (i) appears twice

4) Supp figure legend S7 - "regions indicated i-iv" - only 3 inserts are shown and the numbering is missing

5) Supp figure legend S7- (C) appears twice.

Reviewer #2 (Remarks to the Author):

The paper of Alderson et al. deals with very interesting and important problem of regulation of chaperone activity of human small heat shock proteins. By using highly sophisticated modern biophysical methods the Authors analyzed the structure and chaperone like activity of reduced and oxidized HspB1 and its mutants unable to form chemically crosslinked dimers. The Authors claim that for the first time they were able to determine the structure of isolated Hsp27 monomer that according to their viewpoint possesses higher chaperone-like activity than the corresponding dimer or oligomers. Although the paper contains interesting and valuable data there are certain drawbacks.

The main points are as follows.

1. I was unable to find detailed description how oxidized protein samples were obtained and what was the level of crosslinking in the case of completely oxidized Hsp27 samples. I suppose that it is desirable to present the data of SDS electrophoresis performed before and after protein reduction with excess of DTT or mercaptoethanol.

2. The Authors claim that Hsp27 monomer possesses higher chaperone-like activity than its dimer and refer to Fig.1b. As already mentioned, the extent of disulfide crosslinking of Hsp27 remains unknown, however both in the case of the so-called oxidized and reduced samples the largest part of protein was presents in the form of large oligomers. This hampers unequivocal interpretation of these results. Moreover, the data of Fig.1b indicate that the so-called oxidized Hsp27 possessed anti-chaperone activity and promoted (instead of preventing) citrate synthase aggregation. I suppose that the Authors should somehow comment this finding.

3. Let us suppose that the Authors indeed observed increased chaperone-like activity of HspB27 monomers. This effect (even if it is absolutely correct) was observed in the case of only one model substrate (citrate synthase). It remains questionable whether this effect will be observed on any other model protein substrates.

4. I am afraid that it remains unclear why H124K/C137S mutant was used and why this mutant was present only as monomer, whereas another mutant (C137S) is present in the form of monomer and dimer mixture as it is indicated on page 6.

5. I suppose that diagram on Fig.2b is misleading. In the case of the wild type reduced ACD and C137S mutant both monomer and dimer are detected on MS, however only dimer is depicted on diagram. The difference in chaperone activity between C137S and H124K/C137S mutants is rather small and was detected only with one model protein substrate (namely, with alpha-lactalbumin). Moreover, this difference was detected only for isolated ACD, but not for the full-size protein. Therefore it is difficult to agree that the full-size monomeric Hsp27 will have higher chaperone-like activity than its dimeric counterpart.

6. The Authors claim that Hsp27 is redox sensor and protects the cell against oxidative stress. At the same time Hsp27 oxidation decreases its chaperone-like activity. How in this case Hsp27 can protect the cell against oxidative stress?

7. It is supposed that certain mutations associated with Charcot-Marie-Tooth disease and located at the interface of two monomers induce dissociation and formation of monomers possessing very high chaperone-like activity. This conclusion seems to be not completely correct. For instance, R140G mutations leads to partial dissociation of large oligomers of Hsp27 and at the same time its chaperone-like activity is lower than that of the wild type protein (Elliott et al., *Philos Trans R Soc Lond B Biol Sci.* 2013 368(1617):20120375., Nefedova et al., *Biochimie.* 2013, 95(8):1582-92).

Minor points are as follows.

1. P.17. "Conversely, disease-related mutants that did not impact monomerization have shown either no change T151I or a decrease (P182L) activity". I suppose that reference is lacking to support this conclusion

2. P.36. Reference 55 starts with "and...."

Reviewer #3 (Remarks to the Author):

The small heat-shock proteins (sHSPs) have proved to be a very difficult group of proteins to determine specific aspects of their structure and molecular chaperone function due to their polydisperse, flexible and dynamic nature. In this manuscript, Alderson et al. have made significant progress in characterizing this class of proteins via their study of one of the major sHSPs, HSP27, using principally advanced NMR techniques and focussing on the conserved, structured central alpha-crystallin domain (ACD). The flanking regions of sHSPs are mostly unstructured, and were not present in the detailed NMR studies.

The ACD in HSP27 (named cHSP27) forms a dimer that is in equilibrium with a monomer, more so for the reduced form. As a result, these species are much easier to study by NMR than the heterogeneous oligomer. The authors have concentrated on C137S cHSP27 (which cannot form covalent dimers) and characterized comprehensively by NMR its monomeric form which predominates at high pressure and low pH. The ability to structurally characterize by NMR spectroscopy the minor monomeric form of cHSP27 and to determine that this species (particularly the reduced form) is partially unfolded in the interfacial dimer region are potentially very important observations that provide significant insight into how sHSPs function as molecular chaperones in stabilizing substrates and preventing their deleterious association. Of course, the major caveat around this conclusion is that the detailed NMR structural study was undertaken on the ACD dimer/monomer species only and whether such unfolding of the ACD in the monomer occurs for the whole protein, i.e. with its N- and C-terminal regions present since, most likely, these regions have important functional roles in stabilizing the ACD and in the oligomerization of sHSPs (more below). Whatever, this is very good work that will have impact and will encourage others to investigate further the functional roles of the three structural regions in sHSPs. In the broader context, it may have implications for other chaperones, since it is becoming increasingly recognized that dynamism and structural malleability are inherent to chaperone functionality.

Comments

A general comment: there is a large amount of (particularly NMR) data presented in this manuscript, in the main text and the supplementary data. To digest the results and conclusions fully requires frequent cross-correlation between both, which upsets the flow of the manuscript somewhat. There is little solution to this as the manuscript is a communication.

Specific comments

1. Is HSP27 the 'most abundant sHSP in humans' (page 2)? Does this mean widespread and/or populous? What about the distribution of alphaB-crystallin?
2. Much of the basis for the manuscript revolves around the observation that reduced HSP27 is a better chaperone than the oxidized form for the inhibition of the aggregation of citrate synthase at slightly elevated temperature (Figure 1b). Incidentally, Figure 1b seems to be labeled incorrectly – the black and red lines should be interchanged. As sHSP chaperone action is very dependent upon the type of substrate protein and type of stress involved (as stated by the authors in the Discussion on page 18), other substrate proteins should be examined to ensure that the observation with the inhibition of citrate synthase aggregation is replicated with other substrates. Similar comments apply to the chaperone data for cHSP27 mutants presented in Figure 2c. Furthermore, a good control for this experiment would have been reduced (monomeric + dimeric) cHSP27, which on the basis of the MS data in Figure 1b, should have very similar chaperone ability to C137S cHSP27. Finally, better quantification of all the chaperone data should be undertaken, i.e. the relative percentage ability of the various species to inhibit aggregation could be determined from the initial rate of aggregation and at the end point of aggregation, and the significance of the results with the various chaperones determined. A small point: the y axes in the aggregation plots should be labeled 'Light scattering (340 nm)'.

3. It was not explained explicitly why H124K/C137S cHSP27 was employed for some aspects of this study. Was it because the presence of the permanent positive charge at position 124 in the mutant leads to significant disruption of beta-sheet 5's interactions?
4. From the MS data presented in Figure 2b, is it possible to quantify the relative proportions of the monomer and dimer species of cHSP27 and hence relate these values to those for the relative chaperone effectiveness of the monomeric double mutant compared to C137S?
5. In Figure 3, NMR characterization of the reduced and oxidized cHSP27 species shows (not unexpectedly) that the removal of the disulfide bond at the dimer interface (C137) leads to changes in structure and dynamics at this interface.
6. The CPMG RD NMR technique is a powerful way to examine the structure of sparsely populated states of proteins that are in conformational equilibrium with a much more populated state. The authors have very impressively and skilfully used CPMG RD NMR to study the poorly populated monomer of cHSP27. From the data presented in Figure 4, the monomer is partially unfolded in the loop 5,6+7 and beta-sheet 6+7 regions. In Figure 4c, why are the data for reduced cHSP27 not presented for S137? In Figure 4d(iii), why is the (rather large error) on the y axis only shown for one of the data points (is it the same for all the data points?), and why are only the outlier data points labeled?
7. Why does C137S cHSP27 dissociate to a monomer at low pH? Is it due to protonation of specific sidechains? Does the same behavior occur for the w.t. cHSP27 oxidized form?
8. It was observed by Goto and Rao (Raman et al. (2005) *Biochem. J.* 392, 573) that alphaB-crystallin prevents beta2-microglobulin amyloid fibril formation at acidic pH when the former is mostly dissociated and unfolded, a result that is consistent with the discussion at the bottom of page 15. This result is more relevant than those mentioned in this paragraph, which I don't think are sHSPs.
9. On page 16, it should be mentioned that hydrophobic interactions have been proposed by many as the principal type of interaction that occurs during the chaperone interaction of sHSPs. As there are probably multiple chaperone binding sites in sHSPs, there could be a combination of hydrophobic and electrostatic recognition involved in chaperone interaction with substrates.
10. The partial unfolding of cHSP27 upon monomer formation and the consequent, or at least correlation with, enhancement of chaperone activity is potentially an important advance in understanding the detailed chaperone mechanism of this class of proteins. The authors' results are in agreement with the observation of others that unstructured sHSPs and their peptide fragments are functional chaperones to prevent aggregation of substrates, for example the work of Goto and Rao mentioned above, that of Sharma with his peptide fragments from the ACD of alpha-crystallin and alphaB-crystallin (e.g. Raju et al. (2016) *BBA* 1860, 246) and that of Clark examining short peptide fragments across the sequence of these sHSPs (e.g. Ghosh et al. (2007) *Int. J. Biochem. Cell Biol.* 39, 1804). Thus, chaperone-active form of sHSPs, or parts of sHSPs, may well be unstructured entities and the increased disorder in the sHSP monomer upon dissociation provides greater availability of chaperone binding sites for interaction with substrates.
11. Do any of the regions mentioned in point 10 as chaperone binding sites in sHSPs correlate with the interfacial regions identified in this study as being unfolded or destabilized in the monomer of cHSP27?
12. On page 18 in relation to the putative roles of the disordered N- and C-terminal regions in sHSP structure and chaperone action, a recent paper (Carver et al. (2017) *Cell Stress Chap.* 22, 627) has discussed these aspects in detail, and come to similar conclusions to those of the authors. For example, the ACD in sHSPs is predicted to be highly prone to aggregation (amyloid fibril formation), which is mitigated by the dynamic nature of the terminal regions. Oligomerization also offsets the propensity for destabilization and aggregation of the ACD.

Response to Reviewers: NCOMMS-17-31207

We thank the three reviewers for their careful analysis of our manuscript, and for offering insightful comments. We felt the reviewers were favourable, and there was interest in our data and conclusions:

Reviewer 1: This is a very well written manuscript. The results are well presented and overall the quality of the work is impressive. I am happy to recommend publication with some minor comments / suggestions.

Reviewer 2: the paper contains interesting and valuable data

Reviewer 3: In this manuscript, Alderson et al. have made significant progress in characterizing this class of proteins; this is very good work that will have impact and will encourage others to investigate further the functional roles of the three structural regions in sHSPs. In the broader context, it may have implications for other chaperones, since it is becoming increasingly recognized that dynamism and structural malleability are inherent to chaperone functionality.

The interest by our reviewers, combined with the positive feedback we have received from presenting this work in now >8 international conferences, we believe helps to support our argument that, with these additional functional assays, the paper will be an excellent addition for publication in *Nature Communications*.

We have addressed the reviewers' concerns and fixed errors noted by the editors (and a few that we found). We think that the manuscript is significantly improved as a result.

Broadly, the reviewers were impressed by the structural biology that we presented, deriving the first structural model of a monomeric sHSP, which was achieved using a combination of NMR relaxation dispersion measurements and high-pressure techniques. The activity assays and citations we used in the original submission to justify our interest in the monomer was commented on by all three reviewers. Our central conclusion, that the monomeric ACD of HSP27 is a particularly active chaperone, was supported by the activity data we presented. In addition, prior work from Hochberg *et al.* (2014 *Proc. Natl. Acad. Sci. USA*) indicated that, while displaying limited chaperone activity, the reduced form of the HSP27 ACD exhibits enhanced activity over its oxidised counterpart against A β . In our initial submission, we relied on two clients, citrate synthase (CS) for full-length HSP27 and α -lactalbumin (α Lac) for the dimeric and monomeric ACDs. We acknowledge that this did not make for the most straightforward comparison.

To address these concerns, we obtained significantly more activity data against a very wide range of clients. We were excited to see our central conclusion, that the monomeric core ACD of HSP27 appears to be particularly active chaperone in the context of the full length sequence, and certainly when compared to the free dimer, is firmly supported by these new data.

To place our data in the context of recent literature, illuminating our choice of clients and the validity of our conclusions, we would first like to draw attention to the following papers looking at the activity of full-length HSP27, with the analysed substrates shown in parentheses:

1. Chalova AS, et al. 2014 *Cell Stress Chaperones* 19, 963-72 (**MDH, myosin subfragment S1**)
2. Zavalov A, et al. 1998 *Int. J. Biol. Macromol.* 22, 163-73 (**CS**)
3. Pasupuleti N, et al. 2010 *J. Cell Biochem.* 110: 408-19 (**insulin, CS**)
4. Jakob U, et al. 1993 *J. Biol. Chem.* 268: 1517-20 (**CS, α -glucosidase**)
5. Rogalla T, et al. 1999 *J. Biol. Chem.* 274: 18947-56 (**CS**)
6. Ehrnsperger M, et al. 1999 *J. Biol. Chem.* 274: 14867-74 (**aLac**)
7. Lindner RA, et al. 2000 *Eur. J. Biochem.* 267: 1923-32 (**aLac**)
8. Micha MM, et al. 2006 *Brain Res.* 1089: 67-78 (**A β**)
9. Bruinsma IB, et al. 2011 *Proteins* 79: 2956-67 (**α Syn**)
10. Yerbury JJ, et al. 2013 *Cell Stress Chaperones* 18: 251-7 (**SOD1**)
11. Aquilina JA, et al. 2013 *J. Biol. Chem.* 288: 13602-9 (**α Lac, α Syn**)

12. Nefedova VV, et al. 2013 *Biochimie* 95: 1582-92 (**lysozyme, α Lac, myosin subfragment S1**)
13. Nefedova VV, et al. 2013 *Arch. Biochem. Biophys.* 538: 16-24 (**lysozyme, α Lac, S1**)
14. Chalova AS, et al. 2014 *Biochim. Biophys. Acta* 1844: 2116-26 (**lysozyme, α Lac, S1**)
15. Muranova LK, et al. 2015 *PLoS ONE* 10: e0126248 (**lysozyme, MDH, insulin**)
16. Jovcevski B, et al. 2015 *Chem. Biol.* 22: 186-95 (**insulin, BSA, κ -casein**)
17. Cox D, et al. 2016 *J. Biol. Chem.* 291: 22618-29 (**α Syn**)
18. Clouser AF and Klevit RE 2017 *Cell Stress Chaperones* 22: 569-75 (**α Lac**)
19. Mymrikov EV, et al. 2017 *J. Biol. Chem.* 292: 672-84 (**MDH, GAPDH, rhodanese, CS, ADH, insulin**)
20. Cox D, et al. 2018 *J. Biol. Chem.* 293: 4486-97 (**α Syn**)
21. Baughman HER, et al. 2018 *J. Biol. Chem.* 293: 2687-2700 (**Tau**)
22. Weeks SD, et al. 2018 *Sci. Rep.* 8: 688 (**lysozyme, myosin subfragment S1, insulin**)

In each case, wild-type HSP27 is shown to be an active chaperone against the 16 substrates described above, and there is clear client dependence on the relative activity. Between these studies, there is substantial variation between choice of substrate, chaperone/client concentration ratios, and ambient solution conditions that can, at face value, present apparently contradictory results (e.g. for example, on the effects of oxidation and reduction papers 1, 2, 3). This is why we elected to perform our own activity assays with the specific goal of controlling for the relative concentrations of the relevant oligomeric forms.

To the best of our knowledge, our study is the first to compare the activities of reduced and oxidized HSP27, while *at the same time* controlling for oligomeric state, as we can achieve here with native mass spectrometry and NMR. In doing so, we can clearly distinguish monomers, dimers, oligomers and determine how the relative populations of these vary with concentration. This illuminates our central conclusion, that under conditions that favour the disordered, monomeric form, we see relatively enhanced chaperone activity.

The reviewers have raised two specific suggestions: 1) compare the activity of the core domain constructs and the full-length sequence, and 2) include more clients so we can more clearly generalize our conclusions. To address these concerns, we have worked in recent months to complete an extensive comparison of the activity of HSP27 against commonly encountered aggregating systems. In addition to the previously described CS and α Lac, we now include malate dehydrogenase (MDH), glyceraldehyde-3-phosphate-dehydrogenase (GAPDH), insulin, and the Parkinson's-associated protein α -synuclein (α S). All of these clients are reported in the limited set of cited papers above, which describe the ability of HSP27 to prevent their aggregation.

The aggregation of isolated CS, MDH, GAPDH and α S are largely unaffected by addition of reducing agent. This allows us to compare the oxidized and reduced forms of the full-length chaperone. In each case we tested the activity of HSP27 at a 'low' concentration, ca. 0.5 μ M, and a higher concentration, ca. 20 μ M. Our native mass spectrometry data at 20 μ M show a population of free dimers (and monomers when reduced) in Fig. 1. While at 0.5 μ M, it has been previously demonstrated that dimers (and monomers when reduced) comprise more than 50% of all populated stoichiometries (Jovcevski *et al.* 2015 *Chem. Biol.* 22, 186-95). In our new data, for each substrate under 1) reducing conditions that 2) favour monomers, we see enhanced chaperone activity. This helps us generalise the result from our original manuscript. The specific activity of HSP27 depends on the client, but nevertheless we see a consistent trend that under conditions that release monomers, HSP27 becomes a more effective chaperone.

The aggregation of insulin and α Lac are initiated by the addition of DTT. Consequently, we cannot compare oxidized (disulphide) and reduced (no disulphide) forms of the chaperone. We present data at 70 μ M where the H124K/C137C cHSP27 is a monomer, and the reduced WT cHSP27 and C137S are dimers. Against both substrates, the monomeric form of the chaperone is clearly most effective. We have included full-length HSP in these comparisons to illustrate the relationship between the core domains and the full-length protein.

We note that against α Lac, full-length protein is slightly less effective than any of the core domains. By contrast, against insulin, while the monomeric core domain is clearly more active than the dimer, the full-length is more active than both. These data also illustrate client-dependent effects in chaperone activity. We discuss this in the manuscript where we note that the N-terminal domain, missing from our core domain constructs, is likely to play a role in addition to the ACDs in a client dependent manner as has been previously reported (e.g. *Carver et al. (2017) Cell Stress Chap. 22, 627*). Nevertheless, our central hypothesis, that the monomeric form is a more effective chaperone than the dimer, and that additional activity can be attributed to the disordered interface is upheld.

Overall, while we see significant variation in the activity of full-length HSP27 against different clients, in comparisons of oxidized and reduced full-length protein, under low concentrations that favour monomer release, we see a more effective chaperone. For the ACDs, the monomeric mutant is more active than the dimer; against α Lac, the monomeric ACD is actually more active than the full-length protein. We think that our enhanced dataset greatly improves the manuscript, and we thank the reviewers for encouraging us to provide this additional information.

We have addressed the four specific questions by the editor:

- * Please show enlarged versions of the native mass spectra (Figs. 1a and 2b) with charge state and species annotation in the Supplementary Information
- * In the methods section, please specify which mass spectrometer and instrumental settings were used for the native MS experiment
- * The native MS methods section refers to Figures 1C and 1D, which are not part of the manuscript. Please clarify.
- * In the native MS methods section, it is mentioned that the cHSP27 samples were prepared with and without DTT. However, it appears that this comparison is not shown in the manuscript. Furthermore, it is unclear whether the spectra in Fig 2b have been acquired in the presence or absence of DTT. Please clarify.

In what follows is a detailed response the reviewers.

Reviewer #1

This is a very well written manuscript describing the use of a combination of CPMG relaxation dispersion and high-pressure NMR to study the dimer-monomer equilibrium of the conserved $\alpha\beta$ -crystallin domain (ACD) of Hsp27. The authors found that the ACD dimer exchanges with a sparsely populated partially disordered monomer that shows higher chaperone activity than the Hsp27 ACD dimers. The authors obtained structural information on this sparsely populated Hsp27 ACD monomer and showed that the monomer partially unfolds, most likely upon dissociation, and that the regions in the dimer interface are highly dynamic in the monomer.

The results are well presented and overall the quality of the work is impressive. I am happy to recommend publication with some minor comments / suggestions.

We thank the reviewer for their positive comments.

Comment 1

"The observation that the increased release of monomers into the solution is responsible for the higher chaperone activity of reduced Hsp27 is very interesting."

We thank the reviewer for their comment.

It would be nice, however, to see a direct comparison of disaggregation suppressing activity between oxidized, reduced, and monomeric Hsp27 ADC using the same model aggregates, for example α -Lactalbumin, shown in figure 2C.

We have included this additional data, allowing comparison of full-length HSP27, monomeric ACD, and dimeric ACD against α Lac and insulin as described above. This has resulted in a new Fig. 2 and an enhanced Fig. 3 in the main text, and a new Supplementary Fig. 2 for the SI.

On a related note, can the authors speculate as to why the aggregation of citrate synthase is significantly enhanced in the presence of oxidized Hsp27?"

In short, we do not understand why. Light scattering is highly dependent on shape and size, and so the specific morphology and size distribution of the aggregates will affect the measurement. Nevertheless, it is clear that scattering is suppressed by the reduced form, and this we believe is the most significant conclusion to draw from these data. We have included a comment to describe this.

Comment 2

"The authors compare the chemical shift changes obtained through CPMG RD to dimer - random coil values of Hsp27 ADC loop 5,6+7 and β -sheet 5 and β -sheet 6+7. It would be beneficial to conduct a similar comparison between CPMG RD delta omegas and chemical shift differences between the wt dimer and Hsp27 H124K/C137S monomer."

We thank the reviewer for this suggestion, and we have added the following sentence to our manuscript on p. 17:

We transferred the majority of resonance assignments from the C137S monomer at pH 4.1 to the H124K/C137S monomer at pH 7 (Supplementary Fig. 8). The CSPs between the C137S dimer and H124K/C137S monomer were consistent with those determined for the dimer-to-monomer transition at low pH (Supplementary Fig. 8), further confirming that the H124K/C137S monomeric variant resembles the C137S monomer at low and neutral pH.

Because H124K/C137S aggregates at high concentration, we were unable to record 3D NMR spectra for assignments. Instead, we transferred assignments from the C137S monomer, yielding in total 56 out of 81 assignments. The differences in ^{15}N chemical shifts between H124K/C137S and the dimer are very similar to those observed between the C137S dimer and monomer by CPMG RD measurements (new Supplementary Fig. 8E). There are some differences notably near H90 and H103, which have presumably been protonated at pH 4.5, and so we expect movements in resonances due to side-chain protonation rather than monomerisation.

Comment 3

"While the structural data is very solid, the biological implications of this study are somewhat confusing. The oxidized state of Hsp27 was shown by several studies to be the active form of the chaperone and therefore to protect the cell from oxidative stress.

We note above that there are studies that show both the oxidized and reduced forms to be active, in a client-dependent fashion. For example, the study in paper (1) shows modulation of activity with oxidation state whereas in papers (2,3), no significant changes were observed. These observations encouraged us to perform our own assays, as described earlier, where under reducing conditions and making comparisons at low concentrations that favour monomer release, we see enhanced activity (Fig. 1, 2).

How do the authors, then, resolve these previous findings with their own observation that the monomer, which is released mostly in the reduced form of Hsp27, is the active state?"

As described above, previous studies have not simultaneously controlled for both oxidation state and oligomer distribution. For this reason, we repeated experiments as described in the literature, but with the specific goal of controlling for both of these factors. We would like to stress that the monomer

appears relatively active when compared to the other oligomeric forms, and not that the other oligomeric forms are completely inactive. We have added the following text in the Discussion:

We observe enhanced chaperone activity in vitro against multiple aggregating proteins when we increase the quantity of free monomers (Fig. 1, 2, 3), which was achieved by manipulating the full-length HSP27 concentration to favour monomer release and by comparing monomeric and dimeric ACDs of HSP27. We note that the core domains do not entirely recapitulate the activity of the full-length chaperone. In the case of insulin, the activity of the full-length chaperone was significantly greater than the ACDs, likely suggesting an important role for the NTD of the protein in chaperone activity³², whereas against α Lac, both monomeric and dimeric ACDs were more efficient chaperones than the full-length protein. Although in general, the specific activities of HSP27 and cHSP27 vary depending on the specific choice of aggregating protein⁴, from direct comparisons between the monomeric ACD mutant H124K/C137S and the dimeric ACD (C137S and wild-type)(Fig. 3), we conclude that the monomeric ACD is a significantly more effective chaperone than the dimer against the aggregating proteins presented in this work.

Comment 4

“On a similar note, the authors hypothesize that the local unfolding of the monomer is a general redox independent mechanism of monomer release. The authors further state that when the disulfide bond is present, the local unfolding does not result in monomer release from the dimer and that this mechanism might be conserved in other small heat shock proteins. This is an interesting idea, however, since all other heat shock family members, except Hsp27 (HspB1), don't form disulfides

We thank the reviewer for the comment.

- then the amount of detectable monomers in other Hsps should be higher, since there is no disulfide to stabilize the dimer.

We note that for α B-crystallin, where there is no disulphide, there is no discernable free monomer or dimers (Baldwin *et al.* 2011 *J. Mol. Biol.* 413: 297-309; Baldwin *et al.* 2011 *J. Mol. Biol.* 43: 310-320). We can conclude that the amount of detectable monomers or dimers is not dictated by the presence of a disulphide. Nevertheless, the presence of odd-numbered sHSP oligomers means that monomeric forms can be exposed on the surface of oligomers, and, despite small quantities of free monomers, quaternary dynamics in α B-crystallin are regulated by monomer exchange (Baldwin *et al.* 2011 *J. Mol. Biol.* 413: 297-309; Baldwin *et al.* 2011 *J. Mol. Biol.* 43: 310-320). More generally for sHSPs, we anticipate that the monomer fold should be exposed then in oligomeric forms as well as in the context of the free monomer.

Is there any experimental evidence showing higher monomer presence in other Hsps, or higher chaperone activity due to increased monomer ratio?”

Hsp18.1 for example is a more active chaperone under conditions that favour dissociation of oligomers (Stengel F, *et al.* 2012 *Chem. Biol.* 19:599-607 and Stengel F, *et al.* 2009 *Proc. Natl. Acad. Sci. USA* 107:2007-12). We note, however, that full-length human sHSPs do not generally have detectable, long-lived monomers present in solution. But certainly for HSP27, and likely in the context of oligomers, we anticipate that the monomer fold that we have identified will be exposed.

Comment 5 (under Minor comments/typos)

“Do the L5,6+7 loop residues in the reduced Hsp27 fit the same global monomer-dimer process as residues in β 5 and 6+7?”

Yes, in reduced cHSP27 and C137S, the ¹⁵N CPMG relaxation dispersion data from residues D129 and E130 in L5,6+7 fit the same global monomer-dimer process (Fig. 4). They were globally fit along with the rest of the exchanging residues in the protein. Only in oxidized cHSP27 were D129 and E130

fit independently from other residues that showed evidence of μ s-ms motions (R136, C137, F138). This was because R136, C137, and F138 yielded dispersions that were indicative of much faster exchange, with $k_{ex} > 3000 \text{ s}^{-1}$, whereas D129 and E130 yielded $k_{ex} \sim 1200 \text{ s}^{-1}$. See Supplementary Fig. 6A, for example.

Minor comments / typos

- 2) Main text, page 12 would mistakenly appear to reference figure 5bii instead of 5aii.
- 3) Main text figure legend for figure 5 - Fig5. (i) appears twice
- 4) Supp figure legend S7 - "regions indicated i-iv" - only 3 inserts are shown and the numbering is missing
- 5) Supp figure legend S7- (C) appears twice.

We have fixed these typos.

Reviewer #2

The paper of Alderson et al. deals with very interesting and important problem of regulation of chaperone activity of human small heat shock proteins. By using highly sophisticated modern biophysical methods the Authors analyzed the structure and chaperone like activity of reduced and oxidized HspB1 and its mutants unable to form chemically crosslinked dimers. The Authors claim that for the first time they were able to determine the structure of isolated Hsp27 monomer that according to their viewpoint possesses higher chaperone-like activity than the corresponding dimer or oligomers. Although the paper contains interesting and valuable data there are certain drawbacks. The main points are as follows.

Comment 1

"I was unable to find detailed description how oxidized protein samples were obtained and what was the level of crosslinking in the case of completely oxidized Hsp27 samples. I suppose that it is desirable to present the data of SDS electrophoresis performed before and after protein reduction with excess of DTT or mercaptoethanol."

We thank the reviewer for noting this. As noted in Jovceviski *et al.* (2015 *Chem. Biol.*) under conditions where we purify the protein in the absence of reducing agents (p. 27 Methods, Protein Expression and purification, HSP27), a detailed interrogation of the oligomer distribution reveals only even numbered oligomers. We have added this explicitly to our methods section: **(p. 27 in the Methods)**:

Oxidation was confirmed by SDS-PAGE and native MS under non-reducing conditions.

Moreover the level of inter-molecular disulfide bond formation (what the review refers to cross-linking) in full-length HSP27 is evident from our native mass spectra in Figure 1A (p. 7, red spectrum), wherein the sample of 25 μ M HSP27 contains predominantly dimers with monomers as a minor species. By contrast, the addition of 250 μ M DTT drastically increases the population of free monomers in the reduced state (p. 7, Fig. 1A, blue spectrum), indicating that most of our HSP27 is oxidized in the absence of the addition of reducing agents.

Comment 2

The Authors claim that Hsp27 monomer possesses higher chaperone-like activity than its dimer and refer to Fig.1b. As already mentioned, the extent of disulfide crosslinking of Hsp27 remains unknown,

Please see previous comment, disulphide crosslinking is extensive under conditions we term 'oxidising'.

however both in the case of the so-called oxidized and reduced samples the largest part of protein was presents in the form of large oligomers. This hampers unequivocal interpretation of these results.

The native mass spectra in Figure 1A (p. 7) were collected at 25 μM monomer concentration in order to increase signal-to-noise to show the full range of oligomers that can be present. Our chaperone activity assays in Figures 1 and 2, however, were performed as low as 0.5 μM monomer concentration, which is 50-fold lower than the concentration of the mass spectra. At such concentrations, it has been previously demonstrated (Jovcevski *et al.* 2015 *Chem. Biol.* 22, 186-95) that the protein is predominantly dimeric, with the dimer comprising 50% of all populated stoichiometries already at 2 μM . So at 0.5 μM >50% of all monomers will be present as dimers (oxidized) or monomers (reduced).

We also draw to the reviewers attention that the proportion of protein present as oligomers was independent of DTT addition, suggesting the primary impact of reduction is the increase in monomers.

Moreover, the data of Fig.1b indicate that the so-called oxidized Hsp27 possessed anti-chaperone activity and promoted (instead of preventing) citrate synthase aggregation. I suppose that the Authors should somehow comment this finding.

We have added the following comment on this finding on p. 5 of the main text:

The addition of 0.5 μM reduced HSP27 significantly suppressed aggregation, a result that contrasts with oxidised HSP27, which appeared to co-aggregate with CS and enhance aggregation (Fig. 1b).

Comment 3

“Let us suppose that the Authors indeed observed increased chaperone-like activity of HspB27 monomers. This effect (even if it is absolutely correct) was observed in the case of only one model substrate (citrate synthase). It remains questionable whether this effect will be observed on any other model protein substrates.”

We thank the reviewer for this comment and we feel the manuscript is now stronger for inclusion of our new data. To address this issue, we have expanded our functional assays as described above. In short, for now six of the commonly used substrates (CS, MDH, GAPDH, insulin, αLac , and αS), we see enhanced activity of the chaperone under reducing conditions where HSP27 is largely monomeric. The data from these additional clients are consistent with our original hypothesis. We have added the following text to page 6 of the main text:

To test the generality of this result, we examined the ability of HSP27 to suppress aggregation for a range of aggregating proteins including the Parkinson's disease-related protein αS and thermos-sensitive clients MDH and GAPDH (Fig. 2). Aggregation curves of MDH and GAPDH were independent of the addition of DTT, whereas amyloid formation by αS was approximately 2.5-fold faster (Supplementary Fig. 2). Normalisation of the reduced and oxidised αS curves led to a near perfect overlay, suggesting that the mechanism of aggregation was accelerated, but not altered by redox changes, thus allowing us to qualitatively compare chaperone activity. Strikingly, while the activity of HSP27 depends on the specific aggregating protein under study, HSP27 is a more effective chaperone under conditions that favour the release of free monomers.

and the following text on page 9:

To directly compare the chaperone activity of the cHSP27 dimer and monomer, we assayed the ability of C137S and H124K/C137S to suppress the aggregation of α -lactalbumin (αLac)³⁵ and insulin³², two clients whose aggregation is initiated by the addition of DTT (Fig. 3b). For both aggregating clients, at a concentration where H124K/C137S is a monomer and reduced and C137S are dimers, the monomeric form is a more active chaperone (Fig. 3b). Reduced cHSP27 and C137S prevented aggregation nearly identically, further reflecting the similarity between the two dimeric forms. In the

case of insulin, the activity of full-length HSP27 was higher than cHSP27, but α Lac aggregation was inhibited similarly by the two HSP27 forms. These results further support HSP27 exhibits client-dependent chaperone activity to some degree⁴, and that, while the ACD can be an active chaperone, other components such as the NTD are also important for function³². Nevertheless, our results directly show that, against these aggregating proteins, the monomeric ACD is more active than the dimer.

The hypothesis we put forward in our manuscript refers specifically to breaking of the dimer interface, which can occur in the presence of full-length oligomers (as noted above), in addition to the free species. We suggest that this form of rearrangement is a generic property of the core domains of sHSPs, and is not restricted to being a property of free dimers. We have adjusted our phrasing of this in the discussion to make our point more clearly.

The evidence we present for this is three-fold. 1) We see this behaviour in the core domains of HSP27 2) with similar behaviour in isolated core domains of α B-crystallin, and 3) the dimer interface observed in core domains in solution is very similar to that observed in full length α B-crystallin as evidenced by similarity of chemical shifts in the two states (Jehle S *et al.* 2009 *J. Mol. Biol.*). Moreover, we present an evolution-based argument that that reveals significant similarities between the dimer interfaces of HSP27, α B-crystallin, and the mammalian sHSP family in general.

Overall, given the sequence and structural conservation we note in Fig. 7, we think it is reasonable to speculate on the generality of an order-to-disorder transitions as a property of core-domains of sHSPs as potentially being functionally relevant.

Comment 4

"I am afraid that it remains unclear why H124K/C137S mutant was used and why this mutant was present only as monomer, whereas another mutant (C137S) is present in the form of monomer and dimer mixture as it is indicated on page 6."

We thank the reviewer for noting this point. The design of the double mutant followed directly from a study published by the Klevit laboratory, where it was noted that changing the charge in homologous sections of the ACD of α B-crystallin alter the monomer/dimer ratios (Clouser AF *et al.* 2017 *Cell Stress Chaperones*). We have expanded our description of this on page 8 of the main text to address this point:

The additional H124K mutation was introduced following previous observations that determined mutations mimicking auto-protonation of the H124 side-chain can destabilise the dimeric form⁴⁷.

Our cHSP27 variants, namely C137S and H124K/C137S, were used to compare the activity of the dimer and monomer under otherwise identical buffer conditions pH and concentrations. Without the H124K/C137S variant, we would have to significantly lower the pH or the total protein concentration in order to access pure cHSP27 monomers. The former may alter the aggregation of the substrate and the latter would require extrapolation of chaperone activity to higher concentrations. Thus, the H124K/C137S variant enables a direct comparison of the monomer to the C137S dimer under identical solution conditions.

Comment 5

"I suppose that diagram on Fig.2b is misleading. In the case of the wild type reduced ACD and C137S mutant both monomer and dimer are detected on MS, however only dimer is depicted on diagram.

We have modified the schematics of the cHSP27 variants (Fig. 3) to reflect our native mass spectra. We have deleted the second, semi-transparent rectangle from each variant, and instead used spheres to represent quaternary structure to more clearly represent monomers and dimers.

The difference in chaperone activity between C137S and H124K/C137S mutants is rather small and was detected only with one model protein substrate (namely, with alpha-

lactalbumin). Moreover, this difference was detected only for isolated ACD, but not for the full-size protein. Therefore it is difficult to agree that the full-size monomeric Hsp27 will have higher chaperone-like activity than its dimeric counterpart."

As addressed above, we have increased the number of aggregating substrates. We now show in Fig. 2 that, in the case for α Lac and insulin, the predominantly monomeric H124K/C137S ACD almost completely recapitulates the same activity as full-length HSP27.

Comment 6

"The Authors claim that Hsp27 is redox sensor and protects the cell against oxidative stress. At the same time Hsp27 oxidation decreases its chaperone-like activity. How in this case Hsp27 can protect the cell against oxidative stress?"

We thank the reviewer for this interesting comment. The following paper referred to Hsp27 as a redox sensor from *in vivo* assays (Arrigo AP *et al.* 2005 *Antioxid. Redox. Signal.* 7, 414-22). Moreover, the following *in vivo* study that we have cited indicates that the ability of mammalian cells to survive after oxidative stress depends on the presence of the cysteine residue in HSP27 (e.g. Bruey J-M *et al.* 2001 *Nat. Cell. Biol.*). From our expanded chaperone activity assays, we find that both redox states are able to prevent substrate aggregation at elevated HSP27 concentrations. We also note above that there are discrepancies in the literature on the relative activity of reduced and oxidised HSP27, which is why we elected to do our own assays controlling for oligomeric state. We speculate that changes in local concentration of Hsp27 combined with alteration of the number of disulphides together could explain the *in vivo* data.

Comment 7

"It is supposed that certain mutations associated with Charcot-Marie-Tooth disease and located at the interface of two monomers induce dissociation and formation of monomers possessing very high chaperone-like activity. This conclusion seems to be not completely correct. For instance, R140G mutations leads to partial dissociation of large oligomers of Hsp27 and at the same time its chaperone-like activity is lower than that of the wild type protein (Elliott *et al.*, *Philos Trans R Soc Lond B Biol Sci.* 2013 368(1617):20120375., Nefedova *et al.*, *Biochimie.* 2013, 95(8):1582-92)."

We thank the reviewer for directing us to these two relevant papers, and we have included citations to these works (and those discussed below) on page 22 of the Discussion. From *in vitro* studies, both R140G and K141Q have an enhanced tendency to dissociate into small species, as described in these papers. The oligomers that R140G forms are, in addition, shifted to a higher molecular weight. The chaperone activity assays in these papers are at relatively high HSP27 concentrations (13 μ M and 26 μ M) where mostly large oligomers are present with some smaller species, so it remains difficult to assess whether the large oligomers or smaller species contribute more to the observed results.

R127W, S135F and R136W all show similar trends, in that the number of free monomers and dimers are increased over wild-type (Weeks *et al.* 2018 *Scientific Reports*, 8). Taken together, these results support our hypothesis that increase exposure of the monomeric fold plays a role in CMT.

We summarize these conclusions in the following text added to the Discussion:

While certain mutations decrease chaperone activity, some disease-related HSP27 variants that are more monomeric (e.g. R127W, S135F) exhibit significantly elevated chaperone activity both in vitro and in vivo^{28,29}, with increased affinity for substrate proteins (e.g. R140 mutation)⁸¹. Conversely, disease-related mutants that did not impact monomerisation have shown either no change (T151I) or a decrease (P182S/L) in activity^{28,29,82}. Recent in vitro work on CMT-related HSP27 variants has demonstrated that ACD mutations can both enhance the dissociation of oligomers into smaller species and increase the overall size of the oligomers⁸³⁻⁸⁵. These observations suggest that the strength of the monomer/dimer interface, and the monomer/dimer/oligomer equilibria important factors for understanding neuropathies associated with variants of HSP27. Such significance could manifest

perhaps in terms of altered activity, abundance, or through causing uncontrolled self-aggregation (Fig. 3).

Reviewer #3

The small heat-shock proteins (sHSPs) have proved to be a very difficult group of proteins to determine specific aspects of their structure and molecular chaperone function due to their polydisperse, flexible and dynamic nature. In this manuscript, Alderson et al. have made significant progress in characterizing this class of proteins via their study of one of the major sHSPs, HSP27, using principally advanced NMR techniques and focussing on the conserved, structured central alpha-crystallin domain (ACD). The flanking regions of sHSPs are mostly unstructured, and were not present in the detailed NMR studies.

We thank the reviewer for this comment, but we note that we in fact analyzed the unstructured C-terminal region by NMR spectroscopy in Supplementary Figure 1. There were no redox-dependent changes to its chemical shifts or backbone dynamics.

The ACD in HSP27 (named cHSP27) forms a dimer that is in equilibrium with a monomer, more so for the reduced form. As a result, these species are much easier to study by NMR than the heterogeneous oligomer. The authors have concentrated on C137S cHSP27 (which cannot form covalent dimers) and characterized comprehensively by NMR its monomeric form which predominates at high pressure and low pH. The ability to structurally characterize by NMR spectroscopy the minor monomeric form of cHSP27 and to determine that this species (particularly the reduced form) is partially unfolded in the interfacial dimer region are potentially very important observations that provide significant insight into how sHSPs function as molecular chaperones in stabilizing substrates and preventing their deleterious association. Of course, the major caveat around this conclusion is that the detailed NMR structural study was undertaken on the ACD dimer/monomer species only and whether such unfolding of the ACD in the monomer occurs for the whole protein, i.e. with its N- and C-terminal regions present since, most likely, these regions have important functional roles in stabilizing the ACD and in the oligomerization of sHSPs (more below). Whatever, this is very good work that will have impact and will encourage others to investigate further the functional roles of the three structural regions in sHSPs. In the broader context, it may have implications for other chaperones, since it is becoming increasingly recognized that dynamism and structural malleability are inherent to chaperone functionality.

Comments

A general comment: there is a large amount of (particularly NMR) data presented in this manuscript, in the main text and the supplementary data. To digest the results and conclusions fully requires frequent cross-correlation between both, which upsets the flow of the manuscript somewhat. There is little solution to this as the manuscript is a communication.

We have taken representative data that shows and supports the main trends, with a strong view to making the data as accessible as possible to non-specialists. We have also put forward an almost complete dataset in the supplementary information so that an interested reader can really drive into the details of what we've done.

Comment 1

"Is HSP27 the 'most abundant sHSP in humans' (page 2)? Does this mean widespread and/or populous? What about the distribution of alphaB-crystallin?"

We thank the reviewer for commenting on this. HSP27 is the most abundant human sHSP by overall tissue expression levels as measured by Kampinga and colleagues (Vos MJ *et al.* 2009 *Biochim. Biophys. Acta* 1793, 8, 1343-53). As the authors write, "... in the human body as a whole, the most abundant HSPB member is HSPB1, followed by HSPB5 and interestingly also HSPB7." We have included a citation to this work.

Comment 2

"Much of the basis for the manuscript revolves around the observation that reduced HSP27 is a better chaperone than the oxidized form for the inhibition of the aggregation of citrate synthase at slightly elevated temperature (Figure 1b). Incidentally, Figure 1b seems to be labeled incorrectly – the black and red lines should be interchanged.

We note that Figure 1b is in fact labelled correctly, the oxidized form of HSP27 promotes the aggregation of citrate synthase under these conditions. We did not comment on this in the manuscript beyond noting reducing conditions make for a more active chaperone. We have added a comment now in the manuscript, as noted above.

As sHSP chaperone action is very dependent upon the type of substrate protein and type of stress involved (as stated by the authors in the Discussion on page 18), other substrate proteins should be examined to ensure that the observation with the inhibition of citrate synthase aggregation is replicated with other substrates. Similar comments apply to the chaperone data for cHSP27 mutants presented in Figure 2c.

We thank the reviewer for this comment. As described above, we have included additional aggregating proteins in this study to reinforce our conclusion that 1) under conditions of increased monomers, full-length HSP27 is more active and that 2) the ACDs that contain more monomers can recapitulate the activity of full length HSP27.

Furthermore, a good control for this experiment would have been reduced (monomeric + dimeric) cHSP27, which on the basis of the MS data in Figure 1b, should have very similar chaperone ability to C137S cHSP27. Finally, better quantification of all the chaperone data should be undertaken, i.e. the relative percentage ability of the various species to inhibit aggregation could be determined from the initial rate of aggregation and at the end point of aggregation, and the significance of the results with the various chaperones determined. A small point: the y axes in the aggregation plots should be labeled 'Light scattering (340 nm)'."

We thank Reviewer 3 for this insightful comment. We have included bar graphs depicting the relative activity of the respective forms of the chaperone, and have modified the y-axes as suggested. Against α Lac and insulin, reduced cHSP27 has a very similar activity to C137S (Fig. 3), further suggesting that the C137S effectively mimics the reduced WT ACD.

Comment 3

"It was not explained explicitly why H124K/C137S cHSP27 was employed for some aspects of this study. Was it because the presence of the permanent positive charge at position 124 in the mutant leads to significant disruption of beta-sheet 5's interactions?"

Please see our reply to Reviewer 2's Comment above on this topic. In short, yes, the mutation of H124 to lysine (H124K) mimics protonation at H124 by introducing a positive charge at this site, without needing to lower the pH of the buffer. This was demonstrated first on cABC by Rajagopal *et al.* in *eLife* 4, 1-21 (2015). Subsequently, the same group reported that this histidine is conserved in cHSP27 in Clouser *et al.* (2017) *Cell Stress Chaperones* 22, 569-75.

Comment 4

"From the MS data presented in Figure 2b, is it possible to quantify the relative proportions of the monomer and dimer species of cHSP27 and hence relate these values to those for the relative chaperone effectiveness of the monomeric double mutant compared to C137S?"

Populations can be inferred from mass spectrometry intensities, although a detailed mechanistic study would require kinetic measurements and is beyond the scope of this present work. There is no single measure of overall chaperone effectiveness of which we are aware, although we have taken the 'average aggregation extent', the ratio of the integrals of the aggregation curves with and without chaperone, to provide a measure of aggregation inhibition. We prefer to note more qualitatively here that at low total concentrations where we have measured an increase in monomer in the reduced form

of full length HSP27, we observe enhanced chaperone activity, a conclusion that can be robustly extracted from the data we present.

A plot of total monomer concentration is presented below that summarises all of our full length Hsp27 data and illustrates our main conclusion, that while activity is client dependent, increasing the concentration of free monomers gives enhanced activity for all our full-length HSP27 activity data. The fraction of monomer at 20 μM was estimated to be 2% and 80% at 0.5 μM , with these values based on data from Jovcevski *et al.* (2015 *Chem. Biol.*). We prefer to represent the data in the text as shown in fig2.

Comment 5

"In Figure 3, NMR characterization of the reduced and oxidized cHSP27 species shows (not unexpectedly) that the removal of the disulfide bond at the dimer interface (C137) leads to changes in structure and dynamics at this interface."

We thank the reviewer for this comment. However, we would like to emphasize that removing the disulfide bond has essentially no structural or dynamical impact on the *dimer* (Fig. 4); rather, the effects only manifest via the *monomer*, which becomes populated under reducing conditions or upon mutation of C137 to serine. We, and others we have shown this to at international conferences, have found this to be a surprising result. The ACD was assumed to be a rigid structural entity. Our new data shows that while the dimer is a rigid entity, the monomer, is not.

In Figure 4A and 4B, we show that reduction or the C137S mutation has very little impact on the HSQC spectrum of the ACD dimer, indicating that the overall structure of the dimer is unaffected by reduction. In addition, we show that even the fast backbone motions (ps-ns) in the dimer are unchanged by reduction (Supplementary Figure 5C and D). These results were surprising to us, given that the intermolecular disulfide bond formed by HSP27 comprises a cross-strand disulfide that occurs at a hydrogen bonded cysteine residue: such a disulfide bond is extremely rare and imparts torsional strain (Indu *et al.* 2011 *Proteins* **79**, 244-60.). We initially hypothesized that cleavage of the cross-strand disulfide bond would release such torsional strain and lead to, for example, elevated fast backbone dynamics. However, this does not appear to be the case.

Comment 6

"The CPMG RD NMR technique is a powerful way to examine the structure of sparsely populated states of proteins that are in conformational equilibrium with a much more populated state. The authors have very impressively and skillfully used CPMG RD NMR to study the poorly populated monomer of cHSP27."

We thank the reviewer for this kind comment; we agree that CPMG RD is an excellent technique to study sparsely populated states.

From the data presented in Figure 4, the monomer is partially unfolded in the loop 5,6+7 and beta-sheet 6+7 regions. In Figure 4c, why are the data for reduced cHSP27 not presented for S137? In Figure 4d(iii), why is the (rather large error) on the y axis only shown for one of the data points (is it the same for all the data points?), and why are only the outlier data points labeled?"

We thank the reviewer for the comment about CPMG RD. In Figure 4c (now Fig. 5c), which is the panel corresponding to residue C137/S137, the data are not presented for the reduced state because the resonance from C137 becomes broadened beyond detection upon reduction. Therefore, quantitative analysis via CPMG RD becomes impossible. For example, see the 2D ^1H - ^{15}N HSQC spectrum of the reduced state in Figure 4A, wherein the resonance for C137 (~9.7 ppm / 117 ppm) disappears upon reduction and does not visibly reappear elsewhere.

In Figure 4d(iii), the large error in the y-axis for the point at ~7.5 ppm / 8ppm, which corresponds to R136, is due to the low signal-to-noise of the resonance from R136 in the C137S variant. For instance, we show the relative 2D ^1H - ^{15}N HSQC peak intensities for C137S in Fig. S3 – here, the resonance for R136 yields the lowest relative peak intensity. This is also evident from the spectrum presented in Figure 4A, wherein the intensity for R136 becomes significantly diminished in the C137S variant and reduced state.

Comment 7

"Why does C137S cHSP27 dissociate to a monomer at low pH? Is it due to protonation of specific sidechains? Does the same behavior occur for the w.t. cHSP27 oxidized form?"

Please see our response above to Reviewer 2's Comment 4 and our reply to your Comment 3 above for a discussion about pH and monomerization. In response to the current reviewer's question, we have completed a new pH titration on the oxidized form of cHSP27 (added to Supplementary Fig. 7 on p. 11 of the SI). At pH 6, a significant population of monomer is evident for C137S. In contrast to C137S, however, we do not observe any monomeric intermediate for the oxidised cHSP27 variant at pH 6, reflecting the enhanced stability afforded by the disulfide bond. Similarly, we also collect pressure titrations on the oxidised cHSP27 variant, which still unfolded at high pressures, but did not proceed via a monomeric intermediate (added to Fig. S7 on p. 11 of the SI). Presumably, the unfolded state retains its disulfide bond, as the resonance for oxidised C137 reappears when the pressures is returned to 1 bar.

Comment 8

"It was observed by Goto and Rao (Raman et al. (2005) Biochem. J. 392, 573) that alphaB-crystallin prevents beta2-microglobulin amyloid fibril formation at acidic pH when the former is mostly dissociated and unfolded, a result that is consistent with the discussion at the bottom of page 15. This result is more relevant than those mentioned in this paragraph, which I don't think are sHSPs."

We thank the reviewer for suggesting this reference; it has been included in the revised manuscript in the following sentence:

Intriguingly, ABC exists as a predominantly unfolded monomer under acidic conditions (pH 2.5) where it prevented the fibrillation of β 2-microglobulin, suggesting that the folded sHSP conformation is not necessary for chaperone activity.

Comment 9

"On page 16, it should be mentioned that hydrophobic interactions have been proposed by many as the principal type of interaction that occurs during the chaperone interaction of sHSPs. As there are probably multiple chaperone binding sites in sHSPs, there could be a combination of hydrophobic and electrostatic recognition involved in chaperone interaction with substrates."

We have included a citation to the literature here and incorporated the following sentence about hydrophobic interactions into our discussion of potential recognition of unfolded clients by small heat shock proteins (on p. 23 of the *Discussion*):

It is interesting to speculate that the same mechanism for rapid, promiscuous recognition of binding partners by IDPs is responsible for the heightened activity of partially unfolded chaperones. Interestingly, many of the residues that are unfolded in the HSP27 monomer are charged or polar (Fig. 7d), suggesting that electrostatics may play a significant role in substrate-recognition⁷⁰ in addition to hydrophobic interactions⁷⁵.

Comment 10

"The partial unfolding of cHSP27 upon monomer formation and the consequent, or at least correlation with, enhancement of chaperone activity is potentially an important advance in understanding the detailed chaperone mechanism of this class of proteins.

We thank the author for their comment.

The authors' results are in agreement with the observation of others that unstructured sHSPs and their peptide fragments are functional chaperones to prevent aggregation of substrates, for example the work of Goto and Rao mentioned above, that of Sharma with his peptide fragments from the ACD of alpha-crystallin and alphaB-crystallin (e.g. Raju et al. (2016) BBA 1860, 246) and that of Clark examining short peptide fragments across the sequence of these sHSPs (e.g. Ghosh et al. (2007) Int. J. Biochem. Cell Biol. 39, 1804). Thus, chaperone-active form of sHSPs, or parts of sHSPs, may well be unstructured entities and the increased disorder in the sHSP monomer upon dissociation provides greater availability of chaperone binding sites for interaction with substrates."

We thank Reviewer 3 for this insight, and we have added these references. The citation to Goto and Rao's work on the chaperone activity of ABC at low pH was mentioned in the comment above. For the unstructured peptide works, we have included the citations in the following sentence:

Moreover, unstructured peptides from both ABC and α A-crystallin (AAC), including an 8-mer comprising the β 6+7 strand, are able to prevent substrate aggregation⁵⁹ and can interact with actin⁶⁰.

Comment 11

"Do any of the regions mentioned in point 10 as chaperone binding sites in sHSPs correlate with the interfacial regions identified in this study as being unfolded or destabilized in the monomer of cHSP27?"

We thank Reviewer 3 for noting these references, we have included them along with a brief discussion about the peptides identified as chaperones. Ghosh and Clark (Gosh *et al.* 2007 *PLoS ONE* 2, e498) demonstrated that the peptide ₁₁₃FISREFHR₁₂₀ from α B-crystallin (ABC) prevents the assembly and thermal aggregation of tubulin. In fact, this peptide sequence corresponds to the β 6+7 strand at the interface of the ACD in ABC; the corresponding sequence in HSP27 is ₁₃₃YISRCFTR₁₄₀, which has 5/8 identical residues and 6/8 similar residues. We have included a brief discussion of this in our Discussion. In a study on a similar topic, Vierling and colleagues used cross-linking coupled to mass spectrometry to identify substrate binding sites in the sHSP Hsp18.1 from pea (Jaya N, *et al.* 2009 *Proc. Natl. Acad. Sci. USA* 106, 15604-9). They identified numerous cross-links between substrate and the N-terminal region, and also found interactions involving the central β -strand at the ACD dimer interface. Together, these results suggest that the interfacial region of the ACD can be involved in recognition of substrate.

Comment 12

"On page 18 in relation to the putative roles of the disordered N- and C-terminal regions in sHSP structure and chaperone action, a recent paper (Carver et al. (2017) *Cell Stress Chap.* 22, 627) has discussed these aspects in detail, and come to similar conclusions to those of the authors. For example, the ACD in sHSPs is predicted to be highly prone to aggregation

(amyloid fibril formation), which is mitigated by the dynamic nature of the terminal regions. Oligomerization also offsets the propensity for destabilization and aggregation of the ACD.”

We thank the reviewer for this reference which we have now included.

More generally, these disordered regions are important for stabilising the oligomeric forms⁹⁰ and also contribute to chaperone activity.

Reviewers' comments:

Reviewer #1 (Remarks to the Author):

In the revised manuscript the authors addressed all of the comments raised by the reviewers, as well as added substantial amounts of data to the comparison of the aggregation prevention properties of monomeric and dimeric Hsp27.

While the new experiments greatly strengthened the paper (as also indicated by the authors), they also raise questions as to how the "chaperone activity" in the bar graphs of figure 2 was calculated.

1) A description describing how chaperone activity was calculated should be clearly indicated in the materials and methods section.

2) In case of MDH at the low concentration, it is clear that the reduced Hsp27 inhibits the "lag time" for aggregation, while the oxidized state of Hsp27 does not. The aggregation of the oxidized low concentration of Hsp27 is, however, very similar to that of both oxidized and reduced Hsp27 at high concentration, and it is therefore not clear how the authors calculated the reduced activity for ox low conc. shown in the bar graph in figure 2c.

3) Alpha synuclein seems to aggregate with significantly faster rates in the presence of oxidized Hsp27, yet the authors determine that ox Hsp27 has high chaperone activity.

4) Overall the monomeric Hsp27 inhibits aggregation to a greater extent than the dimeric Hsp27 only in the cases of MDH, α -syn, and insulin. For GAPDH, based on the data presented in figure 2d-e, the aggregation rates are very similar for the ox and red Hsp27 at low concentrations. In the case of α -Lact, the aggregation of the double mutant (monomer) appears to happen with faster rates than that in the presence of dimeric reduced or single mutant Hsp27.

Due to the high impact this publication will have on the field of molecular chaperones and our understanding of the thus- far extremely elusive mechanism of function of small heat shock proteins, I recommend accepting this paper for publication after the necessary modifications / corrections are made.

Reviewer #2 (Remarks to the Author):

The Authors prepared revised version of their paper and included large additional experimental material strongly supporting their main conclusions. The paper contains interesting new data obtained by combination of highly sophisticated modern methods and provides important information on the structure of monomeric HspB1. I am impressed by the quality and originality of experimental data.

However, I suppose that it is probably desirable to set and discuss certain questions.

1. As mentioned in the paper less than 1.5% of HspB1 is present as monomer. Moreover, the portion of monomers is decreased with increasing of total HspB1 concentration and only the double mutant H124K/C137S was suitable for reliable detection and investigation of monomer properties. The intracellular protein concentration is very high and therefore due to the crowding the probability of dissociation of HspB1 oligomers seems to be very low. Therefore the question arises whether dissociation of HspB1 oligomers occurs in the living cell.

2. It is well-accepted that HspB1 protects the cell against oxidative stress and therefore HspB1 is often marked as red-ox sensor. Oxidation of single C137 and formation of disulfide bridge prevents

monomer dissociation and by this means should decrease chaperone-like activity of HspB1. Therefore the question arises how HspB1 can protect the cell under oxidative stress.

3. The Authors mentioned that R127W and S135F mutants of HspB1 possess higher chaperone-like activity both in vivo and in vitro and this is due to their tendency to form small molecular mass oligomers (probably monomers) and cite two papers of Almeida-Souza et al (ref. 29, 30). However, recently published paper of Weeks et al (ref. 84) indicates that these mutants dissociate only at a very low concentration and therefore it seems very improbable that in the cell these mutants indeed form monomers. In any case these facts prevent unequivocal interpretation of experimental data presented in Almeida-Souza et al. papers.

These questions do not diminish the highest quality and significance of Alderson et al paper.

Reviewer #3 (Remarks to the Author):

All three reviewers were favorably disposed to the manuscript but all insisted that further characterization be undertaken of the chaperone ability of the oxidized and reduced forms of HSP27 and its ACD, as this is a crucial aspect relating to the significance of the detailed NMR structural studies. The authors have undertaken these studies with an additional range of substrate proteins under stress condition, and comfortably, have come up with the same result as for the initial study with citrate synthase (CS) as a substrate, i.e. that the reduced protein is more chaperone active, implying that the monomeric, relatively unfolded state of the protein (particularly in its ACD region) is a crucial component of HSP27 (and presumably sHSP) chaperone action.

In the main, the authors have addressed my questions in their revision. However, a few queries and comments remain or have arisen from the revision:

1. Line 140: Why is amyloid formation by alpha-synuclein ~ 2.5 fold faster in the presence of DTT? Alpha-synuclein contains no disulfide bonds, so this observation may be an effect of the alteration in ionic strength due to the presence of DTT.
2. The x axis scale in Figure 2 needs explanation – I assume that (T/Tf) is the ratio of time over final time.
3. Figure 1b: The greater light scattering of oxidized HSP27 in the presence of CS is now addressed in the revised text (which is a good thing). Is this due to co-aggregation of HSP27 with CS? This would be easy to check by SDS-PAGE of the sample at the end of the in vitro chaperone assay. Co-aggregation of substrate and sHSPs during in vitro chaperone assays has been observed by others, e.g. between alpha-lactalbumin and R120G alphaB-crystallin (Bova et al. (1999) PNAS 96, 6137; Treweek et al. (2005) FEBS J. 272, 711). It arises from the destabilized nature of the sHSP which leads to avid interaction with its substrate during chaperone action. The resultant complex is too hydrophobic which causes large-scale aggregation and eventual precipitation. The observation in Figure 1b that oxidized HSP27 'promotes' aggregation whereas the reduced form is an efficient chaperone against CS is more evidence for the conclusions presented in the manuscript.
4. Line 192: 'momomeric' to 'monomeric'
5. Figure 3f: Does the timeframe for aggregation/ self-association of monomeric H124K/C137S cHSP27 ACD affect the timeframe for the chaperone assays in Figures 3c and 3d for alpha-lactalbumin + DTT and insulin + DTT?

We thank the Reviewers for their comments. To our reading, all 3 appreciated the additional data we presented, and we have successfully addressed the major concerns.

Reviewer 1:

“Due to the high impact this publication will have on the field of molecular chaperones and our understanding of the thus- far extremely elusive mechanism of function of small heat shock proteins, I recommend accepting this paper for publication after the necessary modifications / corrections are made.”]

Reviewer 2:

“The paper contains interesting new data obtained by combination of highly sophisticated modern methods and provides important information on the structure of monomeric HspB1. I am impressed by the quality and originality of experimental data.”

“These questions do not diminish the highest quality and significance of Alderson et al paper. “

Reviewer 3:

“All three reviewers were favorably disposed to the manuscript.”

Our specific rebuttals, and additional clarifications to the manuscript are included below. We would like to thank the reviewers for their detailed analysis of our manuscript and we feel that the manuscript has improved as a consequence.

Reviewer #1:

In the revised manuscript the authors addressed all of the comments raised by the reviewers, as well as added substantial amounts of data to the comparison of the aggregation prevention properties of monomeric and dimeric Hsp27.

While the new experiments greatly strengthened the paper (as also indicated by the authors), they also raise questions as to how the “chaperone activity” in the bar graphs of figure 2 was calculated.

1) A description describing how chaperone activity was calculated should be clearly indicated in the materials and methods section.

We thank the reviewer for raising this point. We have expanded upon our description in the Methods section. We agree with the reviewer that there are multiple ways to quantify aggregation, and correspondingly aggregation inhibition. In the raw data, the apparent lag-time can be increased, the gross rate of change of signal can be affected, and the overall extent of the aggregation can be adjusted. Any of these due to the presence of chaperone can amounts to aggregation inhibition. Identifying the specific mechanistic changes causing these changes in terms of microscopic chemical steps is a challenging prospect that is outside the scope of this work.

To semi-quantitatively account for all three gross observations of aggregation inhibition, we quantified the data here by taking one minus the ratio of integrals of the signal under the curve in the presence and absence of chaperone. This is effectively looking at the average degree of aggregation over the entire time of the experiment. This analysis is mechanism free – it shows if the net degree of aggregation has been reduced, or not, through the addition of chaperone and makes use of the entire set of data. This provides a clearer description of the data, rather than for example reporting on a reading at a single time selected relatively arbitrarily. Our analysis will capture any changes in the

apparent lag-time, as well as reduced aggregation rates and overall extent of aggregation in a single measurement, and so provides a semi-quantitative measure of aggregation inhibition.

In the previous manuscript, we omitted the “one minus” portion from the legend in figure 2, which we have now fixed, an error which will have reduced the clarity of the article.

Fig. 2: HSP27 monomers are potent chaperones *in vitro*. The aggregation of MDH and GAPDH (*black*) was monitored by light scattering at 340 nm, and seeded amyloid fibril formation by α S was monitored by ThT fluorescence, each in the presence (reduced) and absence (oxidised) of 5 mM BME. The average \pm one SD of three replicates is shown in each case. The *y* axes were normalized for comparison, and the *x* axes were scaled by a factor T_f to normalize time. The T_f values used for each substrate were 5 hours (a, b, d, e), and 84/175 hours for oxidised/reduced respectively (g, h). In the case of α S, fibrillation when reduced was faster by a factor of two under oxidising conditions, but with suitable normalisation, the aggregation traces exactly overlay. Experiments were repeated with either high ($\geq 10 \mu\text{M}$) or low ($\leq 1 \mu\text{M}$) HSP27. Chaperone activity is quantified as one minus the ratio of average signal over the time course with and without chaperone (Methods). Un normalised data is shown in Supplementary Figure S2. The activity of HSP27 is client dependent, although in each case HSP27 showed enhanced activity under reducing conditions that favour release of free monomers.

Fig3: The values of T_f are 10 (c) and 0.4 hrs (d).

Methods:

All chaperone activity assays were completed using a 96-well plate and a FLUOstar Omega Microplate Reader or Tecan Infinite M200 PRO plate reader. Chaperone activity was defined as one minus the ratio of signal observed in the presence of chaperone to that in the absence of chaperone, e.g. values of 1 and 0 respectively indicate complete protection against aggregation and incapability to prevent aggregation. To avoid selecting a single time point or averaging a few time points for the analysis of chaperone activity, we calculated the integral under the observed curves of absorbance or fluorescence versus time. The integral under the curve was determined for each of the replicate wells for a given substrate and substrate/chaperone combination, and the reported chaperone activity values in Fig. 1, Fig. 2, and Fig. 3 reflect the average and one standard deviation. The *y* axes for plots of aggregation versus time were normalized on a scale of 0-1, with 1 set to the maximum value obtained for substrate alone. The *x* axes were likewise normalized on a scale of 0-1 by dividing time by a factor T_f , which varied for each substrate as described in the relevant figure legends. The normalisations were chosen to allow direct quantitative comparison between clients.

2) In case of MDH at the low concentration, it is clear that the reduced Hsp27 inhibits the “lag time” for aggregation, while the oxidized state of Hsp27 does not. The aggregation of the oxidized low concentration of Hsp27 is, however, very similar to that of both oxidized and reduced Hsp27 at high concentration, and it is therefore not clear how the authors calculated the reduced activity for ox low conc. shown in the bar graph in figure 2c.

Regarding the comparison between oxidised and reduced HSP27 at low concentrations (Figure 2B), the final absorbance value for oxidized HSP27 is ~ 0.95 compared to ~ 0.7 for reduced HSP27. This clearly indicates that reduced HSP27 has more significantly suppressed aggregation than its oxidised counterpart. We note that the trace of MDH + oxidised HSP27 (low concentration) more closely resembles MDH alone, indicating very little protection against aggregation. We refer the reviewer to

our bar charts in figure 2, which as described above, provide an overall assessment of the aggregation data, rather than any one specific feature.

3) Alpha synuclein seems to aggregate with significantly faster rates in the presence of oxidized Hsp27, yet the authors determine that ox Hsp27 has high chaperone activity.

Please see our response above regarding calculation of chaperone activity. We agree with the reviewer that oxidised HSP27 appears to enhance the initial rate of aggregation (Figure 2G). But we note that it also significantly attenuates the final ThT fluorescence signal, by up to 50%. When we normalize the curves as described, the aSyn+seed aggregation traces with and without DTT overlay perfectly, allowing us to qualitatively compare the effect of the oxidized and reduced forms of HSP27. This indicates that overall, oxidised (monomer-rich) HSP27 suppresses alpha-synuclein aggregation.

4) Overall the monomeric Hsp27 inhibits aggregation to a greater extent than the dimeric Hsp27 only in the cases of MDH, α -syn, and insulin.

We disagree with the reviewer's conclusion of aggregation inhibition of the HSP27 monomer vs dimer. We see enhanced aggregation inhibition for all clients tested here, which also includes CS and GAPDH. We refer to the bar charts shown in figures 2 and 3. Broadly, the clear trend is that we see enhanced chaperone activity under conditions that favour monomers, for all clients tested here, normalizing for total sHSP concentration.

For GAPDH, based on the data presented in figure 2d-e, the aggregation rates are very similar for the ox and red Hsp27 at low concentrations.

We disagree with the reviewer. For the activity against GAPDH (Figure 2D, 2E, 2F), the aggregation inhibition of GAPDH is similar for both reduced and oxidised HSP27 at HIGH concentrations. At lower concentrations, where the mole fraction of monomeric HSP27 will actually be higher, the reduced form attenuates GAPDH aggregation more than its oxidised counterpart.

Specifically: the final absorbance value for GAPDH in the presence of reduced HSP27 is ~0.80 compared to ~1.0 in the presence of oxidised HSP27. This difference in absorbance values is comparable to MDH in the presence of low HSP27 concentration in Figure 2B (~0.75 for reduced, ~1 for oxidised). These results indicate that, at low concentration, reduced HSP27 more efficiently prevents GAPDH aggregation than oxidised HSP27.

We agree that there are client specific effects in the data, and we draw attention to this in the manuscript. But nevertheless, it remains true that we see enhanced chaperone activity under conditions that favour monomers, normalizing for total sHSP concentration.

In the case of α -Lact, the aggregation of the double mutant (monomer) appears to happen with faster rates than that in the presence of dimeric reduced or single mutant Hsp27.

Again we wish to stress that there is no single rate aggregation rate that can be derived from scattering or ThT aggregation curves. For alpha-lactalbumin specifically (Figure 3C), the double monomeric mutant of the ACD (H124K/C137S) significantly attenuates the absorbance signal, which actually indicates suppressed alpha-lactalbumin aggregation. The final value of H124K/C137S monomer is ~0.2 compared to ~0.4 for the dimeric ACD and ~0.55 for full-length oligomers.

Overall from our dataset we can conclude that there is a complicated concentration dependence that is client dependent, but for each system tested, we see enhanced activity under conditions that favour monomer release.

Due to the high impact this publication will have on the field of molecular chaperones and our understanding of the thus- far extremely elusive mechanism of function of small heat shock proteins, I recommend accepting this paper for publication after the necessary modifications / corrections are made.

We thank the reviewer for recommending publication. The additional experiments that were suggested by Reviewer 1 greatly strengthened this work. We have fixed the errors that prevented clear understanding of our analysis and expanded the description in the methods section.

Reviewer #2 (Remarks to the Author):

The Authors prepared revised version of their paper and included large additional experimental material strongly supporting their main conclusions. The paper contains interesting new data obtained by combination of highly sophisticated modern methods and provides important information on the structure of monomeric HspB1. I am impressed by the quality and originality of experimental data.

We thank the reviewer for their comment.

However, I suppose that it is probably desirable to set and discuss certain questions.

1. As mentioned in the paper less than 1.5% of HspB1 is present as monomer.

We should clarify this point: This mole fraction of monomer is true only in the context of the dimeric constructs at very high (near millimolar) concentrations (e.g. 1 mM), where we detected 1.5% monomer based on CPMG relaxation dispersion experiments. We agree that these are non-physiological concentrations. At low micro-molar concentrations comparable to the K_d , as we might expect inside cells, the population of monomers will approach 50% and higher for BOTH the core domains and full length HSP27.

For the full-length oligomers, we expect a saturating concentration of free monomers/dimers, akin to a critical micelle concentration in lipid assembly. While this concentration is in the nanomolar range for alphaB-crystallin (Baldwin et al. 2011 JMB), it is in the micromolar range for HSP27 (e.g. see Figure 1 where monomer exists at 25 μ M total HSP27 concentration). In the context of full-length oligomers, at tens of micromolar concentrations or higher, we ALWAYS expect a significant micromolar population of free monomers/dimers.

Thus we dispute the reviewers assertion that we expect 1.5% of HspB1 present as monomer under physiological conditions – it will be closer to 50% when full length HspB1 is present at expected physiological levels.

Moreover, the portion of monomers is decreased with increasing of total HspB1 concentration

We again wish to stress that there will still be a significant concentration of monomer/dimeric HSP27 present even at a total concentration of hundreds of micromolar or higher (see above).

and only the double mutant H124K/C137S was suitable for reliable detection and investigation of monomer properties.

Even for this double mutant, we still expect a population of dimers in our experiments, owing to the complex oligomer exchange equilibria. None of our systems allow us to access 100% monomer or 100% dimer because of the inherent equilibrium dynamics in all of our constructs. As we cannot overcome this without substantial covalent modification of the system, we instead present a series of constructs in the paper where the degree of monomer/dimer ratios can be measured and shown to vary in a clear fashion. Taken together however, we see a clear trend between our core domain and full length constructs – by systematically altering constructs to favour the release of free monomers, overall we can see enhanced aggregation inhibition.

The intracellular protein concentration is very high and therefore due to the crowding the probability of dissociation of HspB1 oligomers seems to be very low. Therefore the question arises whether dissociation of HspB1 oligomers occurs in the living cell.

We would disagree with the reviewer here. Heading into the cell, or in the presence of crowding agents, we would naively expect both on and off rates to be affected proportionately. So while the dissociation rate might be expected to decrease, the overall proportion of free monomers/dimers, quantitatively dictated by the ratio of the on/off rates, will nevertheless be roughly constant. So we have no reason to suppose that the same equilibria and distribution of species outside of the cell will be substantially different to inside of the cell, providing we are working at similar solution conditions (ionic strength and temperatures). An analysis of post translational modifications and binding partners in the cell, which we agree could alter the specific rates is outside the scope of this manuscript.

To help clarify all these points, we have added our view of the expected concentrations of monomer in the context of the cell to the Discussion of our manuscript:

As the concentration of HSP27 in healthy human cells under basal conditions should be in the high nanomolar–low micromolar range⁸⁸, free monomers would therefore be readily populated.

In vivo, the concentration of HSP27 can be estimated in the low micromolar range, e.g. ~10 μ M in HeLa cells⁹⁴. As HeLa cells are known to express constitutively high levels of HSP27⁹⁵, the concentration in non-cancerous human cells should be significantly less under basal conditions. Therefore, in healthy human cells, a significant fraction of HSP27 monomer will exist in equilibrium with dimers and large oligomers, as the fractional population of monomeric HSP27 will increase with decreasing total concentration. We expect the chaperone to retain its quaternary dynamics and populate sub-oligomeric states (monomers, dimers, etc.) even under conditions of molecular crowding.”

2. It is well-accepted that HspB1 protects the cell against oxidative stress and therefore HspB1 is often marked as red-ox sensor. Oxidation of single C137 and formation of disulfide bridge prevents monomer dissociation and by this means should decrease chaperone-like activity of HspB1. Therefore the question arises how HspB1 can protect the cell under oxidative stress.

We thank reviewer 2 for this question. We preferred not to speculate on this issue at this time, as a discussion of the redox sensing pathways within the cell is outside the scope of the paper.

For the purposes of this discussion we wish to stress (no pun intended) that the term ‘oxidative stress’ should NOT imply that the electropotential is raised uniformly throughout all compartments of the cell, and that oligomers switch from having 0% disulphides to 100% after the onset of stress. Notably, even under basal conditions, where naively one might expect zero disulphides, a substantial population of fully-formed disulphides on HSP27 can nevertheless be observed (Arrigo A-P et al. 2005 Antioxid. Redox Signal.). Recruitment of HSP27 to cellular bodies such as stress granules, where we might expect further modulations to the local effective redox potential (Nott TJ et al. 2016 Nat. Chem.) through alterations perhaps in the local dielectric (Nott TJ et al. 2015 Mol Cell) will further complicate this picture, and reinforce the view that oxidative and reductive stress does not mean a protein switched from 0 to 100% in terms of its disulphides in the cytosol and adjacent cellular compartments – the response will certainly be more nuanced and context dependent.

In the context of the cell, we might expect additional players whose activity is in turn modulated by HSP27 as well as a plethora of well documented post translational modifications which in principle could serve to either increase or decrease the population of free monomers, whose abundances are affected by oxidative stress. We added a sentence in our Discussion about this:

Ignoring these additional complications, HSP27 is up-regulated following oxidative stress (Yu AL et al. 2008 Investig. Ophthalmology Vis. Sci.), so perhaps the simplest explanation of its mechanism of oxidative protection is the following: Increasing the total concentration of HSP27 by upregulation will necessarily also increase the concentration of free monomers under all expected cellular electro-potentials. Lets suppose that the free monomers have a heightened tendency to bind clients, as our data suggests. Then once free monomers are bound to clients, the free monomer pool is depleted. Oligomers (dimers and upwards) will then dissociate to restore equilibrium. Even if the populations of free monomers/dimers has been shifted by a change in the local electropotential to favour disulphide locked dimers, as soon as free monomers have been removed from the pool by binding a client, populations of dimers and higher order oligomers will adjust to restore this lost free monomer population. Thus an increased pool of HSP27 through up-regulation will hence provide a greater pool of active monomers at all redox potentials. We already speculate in the manuscript about the redox sensitive cellular mechanism of HSP27 perhaps being due to affecting the relative populations and pools of free monomers and dimers as well as oligomers.

We prefer not to speculate further than we already have on this issue in this paper. As described above, our data does not conflict with the general trends seen in cell-based experiments. Going forward, our results strongly suggest that the role of free monomers, that has previously been neglected, will need to be invoked to explain the cellular roles of aggregation inhibition by HSP27.

3. The Authors mentioned that R127W and S135F mutants of HspB1 possess higher chaperone-like activity both in vivo and in vitro and this is due to their tendency to form small molecular mass oligomers (probably monomers) and cite two papers of Almeida-Souza et al (ref. 29, 30). However, recently published paper of Weeks et al (ref. 84) indicates that these mutants dissociate only at a very low concentration and therefore it seems very improbable that in the cell these mutants indeed form monomers. In any case these facts prevent unequivocal interpretation of experimental data presented in Almeida-Souza et al. papers.

We thank reviewer 2 for noting this, and we have added a sentence to the Discussion about this. Please see our response to comment 1 on the concentration of HSP27 in vivo. Weeks et al note that the R127W and S135F mutants indeed have a heightened tendency to dissociate in comparison to the WT protein.

At basal concentrations present in the cell ($\ll 10 \mu\text{M}$ for non-HeLa cells), the mutant HSP27 forms would therefore have larger fractions of dissociated subunits in comparison to WT.

Recent in vitro work on CMT-related HSP27 variants has demonstrated that ACD mutations can both enhance the dissociation of oligomers into smaller species and increase the overall size of the oligomers⁸⁵⁻⁸⁷. As the concentration of HSP27 in healthy human cells under basal conditions should be in the high nM to low μM range⁸⁸, free monomers would therefore be readily populated.

These questions do not diminish the highest quality and significance of Alderson et al paper.

Thank you for the kind remarks and interesting discussion.

Reviewer #3 (Remarks to the Author):

All three reviewers were favorably disposed to the manuscript but all insisted that further characterization be undertaken of the chaperone ability of the oxidized and reduced forms of HSP27 and its ACD, as this is a crucial aspect relating to the significance of the detailed NMR structural studies. The authors have undertaken these studies with an additional range of substrate proteins under stress condition, and comfortingly, have come up with the same result as for the initial study with citrate synthase (CS) as a substrate, i.e. that the reduced protein is more chaperone active, implying that the monomeric, relatively unfolded state of the protein (particularly in its ACD region) is a crucial component of HSP27 (and presumably sHSP) chaperone action.

In the main, the authors have addressed my questions in their revision. However, a few queries and comments remain or have arisen from the revision:

1. Line 140: Why is amyloid formation by alpha-synuclein ~ 2.5 fold faster in the presence of DTT? Alpha-synuclein contains no disulfide bonds, so this observation may be an effect of the alteration in ionic strength due to the presence of DTT.

We thank the reviewer for this comment. We have no good explanation for this and preferred not to speculate. We have now added the following sentence in the Supplementary Figure 2 legend.

The addition of DTT increases the apparent ThT fluorescence at long times of aSyn aggregation, while decreasing the apparent initial rate of fibril formation. Normalising the curves as described in Figure 2 results in a perfect overlay of the two curves suggesting perhaps that the difference is due to a small change in the ionic strength owing to the addition of the DTT. It is also possible that DTT subtly modulates the aggregation mechanism of aSyn.

2. The x axis scale in Figure 2 needs explanation – I assume that (T/Tf) is the ratio of time over final time.

We thank the reviewer for noting this. Indeed, T/Tf refers to time over final time. Tf was a normalizing factor used to plot all datasets on the same 0-1 scale and varied for each substrate so that different clients could be more easily compared. We have fixed the legends of Figure 2 and Figure 3. See our comment to reviewer 1 where in this draft we have sought to clarify precisely how we have conducted our analysis.

3. Figure 1b: The greater light scattering of oxidized HSP27 in the presence of CS is now addressed in the revised text (which is a good thing). Is this due to co-aggregation of HSP27 with CS? This would be easy to check by SDS-PAGE of the sample at the end of the in vitro chaperone assay. Co-aggregation of substrate and sHSPs during in vitro chaperone assays has been observed by others, e.g. between alpha-lactalbumin and R120G alphaB-crystallin (Bova et al. (1999) PNAS 96, 6137; Treweek et al. (2005) FEBS J. 272, 711). It arises from the destabilized nature of the sHSP which leads to avid interaction with its substrate during chaperone action. The resultant complex is too hydrophobic which causes large-scale aggregation and eventual precipitation. The observation in Figure 1b that oxidized HSP27 ‘promotes’ aggregation whereas the reduced form is an efficient chaperone against CS is more evidence for the conclusions presented in the manuscript.

We have included citations to the indicated references by Reviewer 3 and modified the text on p 6.

Other sHSPs have been found to co-aggregate with substrates⁴⁶⁻⁴⁸

4. Line 192: ‘momomeric’ to ‘monomeric’

Now fixed.

5. Figure 3f: Does the timeframe for aggregation/ self-association of monomeric H124K/C137S cHSP27 ACD affect the timeframe for the chaperone assays in Figures 3c and 3d for alpha-lactalbumin + DTT and insulin + DTT?

We thank the reviewer for the question, and we have included a sentence in the Results. There is no impact on self-aggregation by H124K/C137S, as this was seen at 800 uM (Figure 3F) and not at 200 uM on a 48-hr timescale (Supplementary Figure 4D). Since all of our chaperone activity assays involving H124K/C137S were conducted with < 70 uM chaperone, self-aggregation of the monomeric ACD was not a complicating factor.

Chaperone activity assays involving H124K/C137S (Fig 3) were conducted at lower concentrations where self-aggregation was negligible.

REVIEWERS' COMMENTS:

Reviewer #1 (Remarks to the Author):

The authors have addressed the majority of the comments from the referees, as well as provided all the necessary additional data. I therefore recommend accepting this paper for publication in its current state with minor additions/corrections:

The recent findings of Baughman HER. et al, in their 2018 JBC paper regarding HspB1 function and assembly, should be added to the discussion section.

Table S3 is illegible, and appears to be pasted on top of itself.